

# Near horizon dynamics of three dimensional black holes

Daniel Grumiller[1*] and Wout Merbis[2]

**1** Institute for Theoretical Physics, TU Wien,
Wiedner Hauptstr. 8-10/136, A-1040 Vienna, Austria
**2** Université Libre de Bruxelles & International Solvay Institutes,
Physique Théorique et Mathématique, Campus Plaine - CP 231, B-1050 Bruxelles, Belgium

⋆ grumil@hep.itp.tuwien.ac.at

## Abstract

We perform the Hamiltonian reduction of three dimensional Einstein gravity with negative cosmological constant under constraints imposed by near horizon boundary conditions. The theory reduces to a Floreanini–Jackiw type scalar field theory on the horizon, where the scalar zero modes capture the global black hole charges. The near horizon Hamiltonian is a total derivative term, which explains the softness of all oscillator modes of the scalar field. We find also a (Korteweg–de Vries) hierarchy of modified boundary conditions that we use to lift the degeneracy of the soft hair excitations on the horizon.

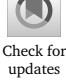
# 1  Introduction

Gravity in the presence of asymptotic boundaries has led to numerous surprises and insights, starting from Bondi, van der Burgh, Metzner and Sachs' seminal work in the 1960ies which proved that the asymptotically flat limit of general relativity does not reduce to special relativity but has an infinite symmetry enhancement due to supertranslations [1, 2]. Brown and Henneaux's discovery in the 1980ies showed that the canonical realization of the asymptotic symmetries in $AdS_3$ Einstein gravity leads to a classical central extension [3], thereby providing a precursor of the $AdS_3/CFT_2$ correspondence. Since then, their asymptotic analysis has been generalized e.g. to higher spins [4,5], lower spins [6] and within Einstein gravity [7–10].

A key outcome of all these considerations is the asymptotic symmetry algebra generated by the boundary charges, since the physical phase space falls into representations of this algebra and different physical states are labelled by the values of these charges, see e.g. [11, 12] and references therein. Depending on the precise form of the asymptotic symmetry algebra, it is possible to generate descendants of a state, so-called "edge-state excitations", by acting on it with raising operators. (The presence of edge states is also familiar from Quantum Hall physics [13]; in a gravity context they are often referred to as "boundary gravitons" since the raising operators have a gravitational interpretation as diffeomorphisms that are not pure gauge at the boundary.)

Boundary charges also exist in the presence of finite boundaries, which can arise in two ways: either there is an actual boundary present in the physical system or one introduces a boundary by cutting out some part of spacetime, see for instance [14, 15]. The prototypical example of the latter is to cut out the black hole interior and to replace the black hole by some membrane [16, 17], corresponding to suitable boundary conditions imposed on a (stretched) horizon [18]. Carlip pioneered the suggestion that such an approach might account for the black hole entropy [19]. The idea to use the black hole horizon as boundary thus has a long pre-history, but concrete proposals for precise boundary conditions and symmetry analyses are relatively recent [20–25].

In the present work we focus on consequences of the near horizon boundary conditions proposed in [22] for the boundary theory. More specifically, we perform, discuss, extend and apply the Hamiltonian reduction of the action under constraints imposed by these near horizon boundary conditions, analogously to the asymptotic analysis of Coussaert, Henneaux and van Driel [26].

Here is a summary of our main results. The near horizon boundary action for each chiral sector,

$$S_{\text{NH}}[\Phi_n, \Pi_n] = \int dt \left( \sum_{n \geq 0} \Pi_n \dot{\Phi}_n - H_{\text{NH}} \right) \tag{1}$$

depends on a scalar field $\Phi(t, \sigma) = \sum_{n \in \mathbb{Z}} \Phi_n(t) e^{in\sigma} + \sigma J_0(t)$, the momentum of which essentially is given by its spatial derivative, $\Pi_n(t) = -\frac{k}{2} J_{-n}(t)$ with $J_n(t) = in\Phi_n(t)$ for $n \neq 0$, like in the Floreanini–Jackiw action for self-dual scalars [27]. Remarkably, the near horizon

Hamiltonian depends only on the momentum zero mode $J_0$

$$H_{\text{NH}} = \frac{k}{2}\,\zeta J_0\,. \tag{2}$$

The equations of motion are solved by

$$\Phi(t,\sigma) = \Phi_0(t) + J_0\,\sigma + \sum_{n\neq 0} \frac{J_n}{in}\,e^{in\sigma}\,, \tag{3}$$

where $J_0$ is related to the BTZ horizon radii and $\Phi_0(t) = -\zeta t + \phi$ with constant $\zeta$ and $\phi$. The mode decomposition (3) is reminiscent of the ultrarelativistic/tensionless limit of string theory [28].

The near horizon Hamiltonian density being a total derivative term implies the softness of all near horizon oscillator modes $J_n$, which provides yet another way to see that $J_n$ generate soft hair excitations of black holes [22] in the sense of Hawking, Perry and Strominger [29].

If one desires to attribute black hole entropy to soft hair degeneracy [25, 30] then one of the problems is the infinite degeneracy of soft hair excitations. The naive physical intuition behind a possible resolution is that infinite blueshift factors at the horizon multiplying zero energy excitations could yield a finite result. Producing such a cutoff in a controlled way is essential for applications to black hole entropy, like in the fluff proposal [31,32]. In the present work we find a novel way to generate such a cutoff, by considering our near horizon boundary conditions as limiting case of an analytically continued 1-parameter family of boundary conditions. The zeroth, first and second member of this family yields, respectively, near horizon, Brown–Henneaux and Korteweg–de-Vries (KdV) boundary conditions. The associated boundary Hamiltonian densities for any positive integer $N$ generalize the result (2)

$$\mathcal{H}_N = \frac{k}{4\pi}\,\zeta_N \mathcal{J}^{N+1} + \sum_{i=1}^{N-1} h_{i,N} \mathcal{J}^{N-i-1}\big(\partial_\sigma^i \mathcal{J}\big)^2 + \mathcal{H}_N^{\text{nl}}, \qquad \mathcal{J} = \Phi'\,, \tag{4}$$

which no longer is a total derivative term ($h_{i,N}$ are field-independent coefficients and $\mathcal{H}_N^{\text{nl}}$ vanishes for $N \leq 4$). Setting $N = 0$ in (4) recovers (2). If instead we take the limit $N = \epsilon \to 0^+$ we get a Hamiltonian with log contribution

$$\mathcal{H}_{\text{log}} = \frac{k}{4\pi}\,\zeta_\epsilon \Phi' \ln\big(\Phi'\big) \tag{5}$$

that provides a cutoff on the soft hair spectrum.

This work is organized as follows. In Section 2 we review our near horizon boundary conditions and their relation to Brown–Henneaux boundary conditions. In section 3 we perform the reduction of the Chern–Simons action to a boundary action, up to one boundary term. In section 4 we fix this boundary term explicitly in different ways, corresponding to near horizon, Brown–Henneaux and KdV boundary conditions as well as generalizations thereof. In section 5 we consider an analytic continuation of this KdV hierarchy and focus on the limit when continuously approaching near horizon boundary conditions, in order to make the soft hair excitations slightly non-soft. In section 6 we compare our results with those of the fluff proposal and tensionless strings.

**Note added:** After finishing our work we received a preview [33] that also considers a hierarchy of integrable deformations of the near horizon boundary conditions [22]. The Gardner hierarchy employed in that work contains the KdV hierarchy studied in our section 4.3 as special case (for $b = 0$ in their notation). As their work does not address the boundary actions discussed in the present paper our respective works are complementary.

## 2 Near horizon boundary conditions

In this section we review the near horizon boundary conditions for AdS$_3$ Einstein gravity [22]. We also list our conventions here.

### 2.1 AdS$_3$ gravity in the bulk

Three dimensional Einstein gravity is a topological gauge theory which can be described as a Chern–Simons theory for the appropriate gauge group [34,35][1]. The gauge group reflects the isometries of the maximally symmetric vacuum of the theory, which in turn depends on the sign of the cosmological constant $\Lambda$. In the present work we focus on negative cosmological constant, $\Lambda = -1/\ell^2$, mainly due to the fact that the presence of BTZ black hole solutions [36,37] requires negative $\Lambda$.

AdS$_3$ gravity is described by SO(2,2) $\cong$ SL(2,$\mathbb{R}$)× SL(2,$\mathbb{R}$) Chern–Simons theory. The Einstein–Hilbert action (neglecting boundary terms) is given as

$$S_{\text{EH}}[A,\bar{A}] = S_{\text{CS}}[A] - S_{\text{CS}}[\bar{A}], \tag{6}$$

where

$$S_{\text{CS}}[A] = \frac{k}{4\pi} \int_{\mathcal{M}} \text{tr}\Big(A \wedge dA + \frac{2}{3} A \wedge A \wedge A\Big), \tag{7}$$

and the Chern–Simons level is given by $k = \frac{\ell}{4G}$ ($G$ is Newton's constant). The gauge connections $A, \bar{A}$ take values in the algebra $\mathfrak{sl}(2,\mathbb{R})$ with generators $L_{-1}, L_0, L_{+1}$ and commutators

$$[L_m, L_n] = (m-n)L_{m+n} \qquad m,n = -1,0,+1. \tag{8}$$

The trace in the action (7) is a non-degenerate symmetric bilinear form on the Lie algebra, chosen as

$$\text{tr}(L_1 L_{-1}) = -1 \qquad \text{tr}(L_0^2) = \frac{1}{2} \qquad \text{tr}(L_{\pm1}L_0) = 0 = \text{tr}(L_{\pm1}^2). \tag{9}$$

Sometimes an explicit representation in term of ($2\times2$) matrices is needed. We make a standard choice compatible with the commutation and trace relations above.

$$L_{-1} = \begin{pmatrix} 0 & -1 \\ 0 & 0 \end{pmatrix} \qquad L_0 = \frac{1}{2}\begin{pmatrix} 1 & 0 \\ 0 & -1 \end{pmatrix} \qquad L_{+1} = \begin{pmatrix} 0 & 0 \\ 1 & 0 \end{pmatrix}. \tag{10}$$

The Cartan variables, dreibein and (dualized) spin-connection, are given as linear combinations of the Chern–Simons connections $A, \bar{A}$.

$$e = \frac{\ell}{2}\big(A - \bar{A}\big), \qquad \omega = \frac{1}{2}\big(A + \bar{A}\big). \tag{11}$$

The metric is bilinear in the dreibein and thus also bilinear in the Chern–Simons connections.

$$g_{\mu\nu} = 2\,\text{tr}(e_\mu e_\nu) = \frac{\ell^2}{2}\,\text{tr}\big((A-\bar{A})_\mu(A-\bar{A})_\nu\big). \tag{12}$$

All solutions of three dimensional gravity are locally gauge equivalent to each other, but differ up to boundary terms or global identifications. Hence the specification of boundary conditions is a crucial part of the definition of the theory under consideration. The boundary conditions will determine which gauge transformations are proper gauge transformations, in the sense that they keep the boundary data invariant, and which are improper gauge transformations that turn into symmetry transformations of the boundary theory.

---

[1]The relation between the Chern–Simons and metric formulations of Einstein gravity is not fully understood at the quantum level.

## 2.2 Boundary conditions

Suppose that our manifold $\mathcal{M}$ is topologically an annulus times time, with the outer boundary being a hypersurface of constant large radius approaching the asymptotically AdS-boundary and the inner boundary of the annulus corresponding to a stretched horizon, i.e., a hypersurface of constant radius $r$ close to the locus of a black hole horizon. We equip the manifold with a coordinate system $(t, \sigma, r)$, where $t$ is the time coordinate, $\sigma \sim \sigma + 2\pi$ the coordinate around the annulus and $r$ some radial coordinate so that the asymptotic boundary is at $r \to \infty$ and the inner boundary $\partial \mathcal{M}$ at $r \to 0$.

In section 2.3 we are going to impose boundary conditions inspired by near horizon considerations at the inner boundary. While in principle one could independently impose a different set of boundary conditions at the asymptotic boundary, we choose the same boundary conditions there, which guarantees that the respective boundary conditions are (trivially) consistent with each other. To make sure of this, it is convenient to express the Chern–Simons connection in radial gauge

$$A = b(r)^{-1} \big( \mathrm{d} + a(t, \sigma) \big) b(r) \tag{13}$$

such that $b(r) \in \mathrm{SL}(2, \mathbb{R})$ depends only on the radial coordinate and the boundary connection $a(t, \sigma) = a_t \mathrm{d}t + a_\sigma \mathrm{d}\sigma$ only has legs in the $(t, \sigma)$ plane. Moreover, the group element $b(r)$ is not allowed to vary, $\delta b = 0$. The choice (13) makes it manifest that the limits $r \to \infty$ and $r \to 0$ yield the same sets of boundary conditions. This is so because all the state-dependent information is contained in the boundary connection $a(t, \sigma)$, which is independent from the radial coordinate.

On an equal time slice, the asymptotic charges on the boundary are obtained by functionally integrating

$$\delta Q[\varepsilon] = -\frac{k}{2\pi} \oint \mathrm{d}\sigma \, \mathrm{tr}\big( \varepsilon \, \delta a_\sigma \big) \tag{14}$$

for asymptotic symmetry transformations $\varepsilon$ satisfying $\delta a_\sigma = \partial_\sigma \varepsilon + [a_\sigma, \varepsilon]$ (see e.g. [38]).

Hence $a_\sigma$ contains all information about the asymptotic charges, and specifying boundary conditions means specifying the form of $a_\sigma$ and its allowed fluctuations. By contrast, the time component $a_t$ contains the information about the sources of the boundary theory. It can always be taken to be proportional to an asymptotic symmetry transformation with arbitrary parameter, which would then play the rôle of a chemical potential for the corresponding boundary charge [39].

The interplay between boundary conditions, holonomies of the gauge connection and SL(2) conjugacy classes is reviewed in appendix A.

## 2.3 Near horizon boundary conditions

The boundary conditions of [22] are formulated as the set of diffeomorphisms preserving the near horizon expansion of the non-extremal BTZ black hole. Their purpose was to be able to ask conditional questions given the presence of a black hole in the bulk, so we restrict ourselves to the BTZ black hole subsector of solutions. From the analysis in appendix A we see that in radial gauge this is equivalent to imposing the connections to have hyperbolic holonomies. So we may write them as ($\mathcal{J}^\pm$ are real functions)

$$a_\sigma = \mathcal{J}^+(t, \sigma) L_0, \qquad\qquad \bar{a}_\sigma = -\mathcal{J}^-(t, \sigma) L_0. \tag{15}$$

Under boundary condition preserving gauge transformations they transform as

$$\delta_\varepsilon \mathcal{J}^\pm L_0 = \pm \partial_\sigma \eta^\pm L_0, \tag{16}$$

where $\varepsilon = \eta^+ L_0 + \eta^+_\pm L_{\pm 1}$ and $\bar{\varepsilon} = \eta^- L_0 + \eta^-_\pm L_{\pm 1}$. The gauge parameters $\eta^\pm_\pm$ do not appear in the transformation laws or in the charges below, and hence correspond to proper gauge transformations. The variation of the charges (14) is easily integrated, assuming that $\eta^\pm$ are state-independent

$$Q[\eta^\pm] = -\frac{k}{4\pi} \int d\sigma \, \eta^\pm \mathcal{J}^\pm \,. \tag{17}$$

A Fourier decomposition of the charges with respect to the angular coordinate $\sigma$ yields an asymptotic symmetry algebra consisting of two affine $\hat{u}(1)$ current algebras[2]

$$i\{J_n^\pm, J_m^\pm\} = \pm \frac{2}{k} n \, \delta_{n+m,0}, \qquad \{J_n^\pm, J_m^\mp\} = 0 \,. \tag{18}$$

Here $J_n^\pm = \frac{1}{2\pi} \oint d\sigma \, \mathcal{J}^\pm e^{in\sigma}$ are the Fourier components of $\mathcal{J}^\pm$. (Note the conventional factor $\frac{2}{k}$ relative to [22].) This algebra can be rewritten as a Heisenberg algebra with infinitely many canonical generators $X_n, P_n$ and two Casimirs ($X_0$ and $P_0$) at its center. Hence these near horizon boundary conditions are also referred to as "Heisenberg boundary conditions".

To see that (15) indeed corresponds to black hole solutions we should construct the metrics associated with these boundary conditions. First we write $a_t$ and $\bar{a}_t$ proportional to a gauge transformation with arbitrary, state-independent parameters $\zeta^\pm$.

$$a_t = -\zeta^+ L_0, \qquad \bar{a}_t = -\zeta^- L_0. \tag{19}$$

Then we should find a suitable group element $b$ to construct the metric (12) from (13). The choice of [40] is $b = \exp\left(\frac{r}{2\ell}(L_+ - L_-)\right)$ and $\bar{b} = b^{-1}$. Other choices for $b(r)$ are suitable as long as they lead to a non-degenerate metric which contains as much information as the gauge connections $a, \bar{a}$. This particular choice leads to a metric which, expanded near $r = 0$ (and assuming a co-rotating frame, $\zeta^+ = \zeta^-$) gives Rindler spacetime,

$$ds^2 = -\kappa^2 r^2 \, dt^2 + dr^2 + \frac{\ell^2}{4}(\mathcal{J}^+ + \mathcal{J}^-)^2 \, d\varphi^2 + a(\mathcal{J}^+ - \mathcal{J}^-) r^2 \, dt d\varphi + \dots, \tag{20}$$

with Rindler acceleration $\kappa = \zeta^+ = \zeta^-$. For simplicity henceforth we assume $\zeta^\pm$ to be constant, implying on-shell time-independence of $\mathcal{J}^\pm$ (dot means $\partial_t$),

$$\dot{\mathcal{J}}^\pm = 0 \,. \tag{21}$$

Finally, writing the functions $\mathcal{J}^\pm(\sigma)$ as

$$\mathcal{J}^\pm(\sigma) = \frac{\gamma(\sigma)}{\ell} \pm \omega(\sigma), \tag{22}$$

the full metric constructed from this configuration [using (12)] for constant $\mathcal{J}^\pm$ becomes the BTZ metric with inner and outer horizons $\gamma$ and $\ell|\omega|$,

$$r_+ = \gamma \qquad r_- = \ell|\omega| \,. \tag{23}$$

There are a number of crucial differences as compared to Brown–Henneaux (or other) boundary conditions:

- **Soft Heisenberg hair.** The zero modes $J_0^\pm$ commute with all generators $J_n^\pm$. Thus, non-trivial descendants of some state generated by acting on it with products of $J_n^\pm$ (with negative $n$) have the same $J_0^\pm$ eigenvalues as the original state. Since the near horizon Hamiltonian is given by the sum of these zero modes [22] this means that all such descendants are soft in the sense that they do not change the energy eigenvalue, concurrent

---

[2]We write $i$ times Poisson brackets so that the right hand sides do not change when passing to commutators.

with the proposal of Hawking, Perry and Strominger. By contrast, for Brown–Henneaux boundary conditions Virasoro descendants generated by acting with $L_n^\pm$ (with negative $n$) on a given state will raise the $L_0^\pm$ eigenvalues of such descendants as compared to the original state, and hence descendants have a higher energy (as measured by the Hamiltonian $L_0^+ + L_0^-$) than the state from which they originate.

- **Fixed temperature.** Since Rindler acceleration $a = 2\pi T$ and horizon temperature $T$ are fixed we are automatically in the canonical ensemble. Their state-independence implies that all states in our theory have the same temperature. By contrast, for Brown–Henneaux boundary conditions different BTZ black holes generally have different temperatures.

- **Regularity of excitations.** All soft hair excitations (with real $\mathcal{J}^\pm$) of black holes are compatible with regularity conditions, in particular with the absence of conical defects. Such excited black holes are sometimes referred to as "black flowers". By contrast, for Brown–Henneaux boundary conditions only extremal black holes can carry (Virasoro) excitations without generating singularities [41], and generally only one BTZ black hole is free from conical defects for given temperature and angular rotation.

- **Reducibility parameter.** Gauge transformations that vanish on-shell are called reducibility parameters (see e.g. [42]). In a gravity context typically they do not exist outside of mini-superspace models, since this would amount to vector fields that are Killing for all geometries compatible with a given set of boundary conditions. For our boundary conditions, however, $\partial_t$ is a Killing vector for all geometries, including softly excited ones. Thus, we do have a non-trivial reducibility parameter. By contrast, for Brown–Henneaux boundary conditions there is no vector field that is Killing for all Bañados geometries (118) and hence no reducibility parameter.

- **Abelianization.** Up to the central extension, the near horizon symmetries (18) are abelian, which at a technical level is a direct consequence of the connection (15) residing exclusively in the Cartan subalgebra. By contrast, Brown–Henneaux boundary conditions lead to non-abelian asymptotic symmetries (regardless of central extensions).

Before deriving the boundary action we comment on our terminology of 'near horizon boundary conditions'. Due to our separation (13) of radial and boundary coordinates the charges (14) are independent from the radius and thus can be envisoned to be localized at any $r = $ const. hypersurface, including one near the horizon or, alternatively, one near the asymptotic AdS boundary. Thus, one could also refer to our boundary conditions as 'asymptotic boundary conditions', which is more in line with tradition. However, it is fair to say that from an asymptotic observer's perspective our boundary conditions are somewhat bizarre; as shown in [22, 40] the usual Fefferman–Graham expansion reveals that the sources depend in a very specific, but complicated, way on the charges, with no clear asymptotic interpretation, other than that this leads to a rather simple set of asymptotic symmetries, given by (18). However, as the list of properties above shows from a near horizon observer's perspective the physical meaning of our boundary conditions is very clear and natural: these boundary conditions guarantee that surface gravity is state-independent and that all state-dependent excitations in our theory are compatible with regularity at the horizon. This, together with the soft hair property, seems like a good reason to associate our boundary conditions with a near horizon observer rather than an asymptotic one.

# 3 Reduction of the Chern–Simons action

In this section we reduce the Chern–Simons action to a two dimensional field theory by first reducing AdS$_3$ Einstein gravity to a sum of two chiral Wess–Zumino–Witten (WZW) models and then imposing the boundary conditions as constraints on the WZW currents. We shall also discuss the presence of non-trivial holonomies in the bulk. We consider here the reduction of Chern–Simons theory to a single boundary of the annulus.

## 3.1 General aspects of the Hamiltonian reduction

Following [43] we rewrite the two Chern–Simons actions in (6) as two WZW actions for SL(2, $\mathbb{R}$). At this stage, the reduction is very similar to the one with asymptotically AdS boundary conditions first observed in [26] (see also [44] for a review). It is most easily obtained after a Hamiltonian decomposition of the action (6). We focus in the rest of this work on one chiral sector ($S_{\text{CS}}[A]$) and drop ±-superscripts, as the barred sector is analogous.

The Hamiltonian form of the action (7), supplemented by a boundary term $I_{\text{bdy}}$,

$$S_{\text{CS}}[A] = \frac{k}{4\pi} \int_{\mathcal{M}} \mathrm{d}t\mathrm{d}r\mathrm{d}\sigma \, \mathrm{tr}\big(A_r \dot{A}_\sigma - A_\sigma \dot{A}_r + 2A_t F_{\sigma r}\big) + I_{\text{bdy}} \tag{24}$$

is our starting point for the reduction. The boundary term $I_{\text{bdy}}$ will be fixed such that the variational principle is well-defined, i.e., the first variation of the action

$$\delta S_{\text{CS}}[A]\big|_{\text{EOM}} = \delta I_{\text{bdy}} - \frac{k}{2\pi} \int_{\partial\mathcal{M}} \mathrm{d}t\mathrm{d}\sigma \, \mathrm{tr}\big(A_t \delta A_\sigma\big) = \delta I_{\text{bdy}} - \frac{k}{2\pi} \int_{\partial\mathcal{M}} \mathrm{d}t\mathrm{d}\sigma \, \mathrm{tr}\big(a_t \delta a_\sigma\big) \tag{25}$$

vanishes on-shell. In the second equality we used the decomposition (13), assuming again state-independence of the group element $b$, i.e., $\delta b = 0$. We are going to fix $I_{\text{bdy}}$ in the next section and focus on the symplectic terms in (24) in the remainder of this section.

The constraint $F_{\sigma r} = 0$ is solved locally by

$$A_i = G^{-1}\partial_i G \qquad i = \sigma, r \qquad G \in \text{SL}(2, \mathbb{R}). \tag{26}$$

Globally, there may (and will) be holonomies in the gauge connection. There are two ways to treat them. One may write the gauge connection as sum of a periodic group element $g$ plus a term representing the holonomy. Alternatively, the holonomies can be encoded in the periodicity properties of the group element $g$. We follow the latter approach and write

$$G(t, r, \sigma + 2\pi) = h(t)G(t, r, \sigma), \tag{27}$$

where $h \in \text{SL}(2, \mathbb{R})$, with $\mathrm{tr}(h) = H_\sigma(a_\sigma)$ at the boundary, using the holonomy definition (111). We assume in this work that $h$ depends only on time.

With the assumptions above the action (24) decomposes into two boundary actions; one at the $r$-boundary and a contribution at the $\sigma$-boundary

$$S_{\text{CS}}[A] = \frac{k}{4\pi} \int_{\mathcal{M}} d^3x \, \mathrm{tr}\big(\partial_\sigma(G^{-1}\partial_r G G^{-1}\partial_t G) - \partial_r(G^{-1}\partial_\sigma G G^{-1}\partial_t G)\big) \tag{28}$$
$$- I_{\text{WZ}}[G] + I_{\text{bdy}},$$

with the Wess–Zumino (WZ) term

$$I_{\text{WZ}}[G] = \frac{k}{12\pi} \int_{\mathcal{M}} \mathrm{tr}\big(G^{-1}\mathrm{d}G\big)^3. \tag{29}$$

The Wess-Zumino term evaluates to a total derivative as well and hence the action (28) solely consists out of boundary contributions. In the remainder of this section we choose $G(t, r, \sigma)|_{\partial\mathcal{M}} = g(t, \sigma)$. This implies $b = 1$ at the boundary, which holds particularly for the choice of $b$ two lines below (19).

## 3.2 Near horizon action

To implement the boundary conditions described in section 2.3 it is convenient to perform a Gauss decomposition of the $SL(2, \mathbb{R})$ group element in the action (28)

$$G = e^{XL_+} e^{\Phi L_0} e^{YL_-}, \tag{30}$$

where $\Phi, X, Y$ are fields depending on all spacetime coordinates, while their pullback to the boundary is $r$-independent since $G|_{\partial \mathcal{M}} = g$. In terms of these fields, the action (28) splits into two parts

$$S[A] = S[\Phi, X, Y] = I_{r-\text{bdy}}[\Phi, X, Y] + I_{\sigma-\text{bdy}}[\Phi, X, Y], \tag{31}$$

with

$$I_{r-\text{bdy}}[\Phi, X, Y] = \frac{k}{4\pi} \int_{\partial \mathcal{M}} dt d\sigma \left( \frac{1}{2} \dot{\Phi} \Phi' - 2e^{\Phi} X' \dot{Y} \right) + I_{\text{bdy}}. \tag{32}$$

The $\sigma$-boundary terms receive contributions from the first term in (28) and from the Wess-Zumino term (29). It reads

$$I_{\sigma-\text{bdy}} = \frac{k}{4\pi} \int d^3x \, \partial_\sigma \left[ \frac{1}{2} \partial_r \Phi \dot{\Phi} - 2e^{\Phi} \partial_r X \dot{Y} \right]. \tag{33}$$

We can simplify the $\sigma$ boundary term by using the periodicity conditions imposed by the holonomy. For our near horizon boundary conditions, the holonomy can be encoded by taking

$$h(t) = \exp(2\pi J_0(t) L_0), \tag{34}$$

where $J_0$ is defined as the (suitably normalized) zero mode of $\mathcal{J}$.

$$J_0(t) = \frac{1}{2\pi} \oint d\sigma \, \mathcal{J}(t, \sigma). \tag{35}$$

The periodicity property (27) implies periodicity conditions on the fields $\Phi, X$ and $Y$ appearing in the Gauss decomposition

$$\Phi(t, \sigma + 2\pi) = \Phi(t, \sigma) + 2\pi J_0(t), \tag{36a}$$

$$X(t, \sigma + 2\pi) = e^{-2\pi J_0(t)} X(t, \sigma), \tag{36b}$$

$$Y(t, \sigma + 2\pi) = Y(t, \sigma). \tag{36c}$$

This implies that the $\sigma$ boundary term evaluates to a total $r$-derivative

$$I_{\sigma-\text{bdy}} = \frac{k}{4} \int dr dt \, \partial_r \Phi|_{\sigma=0} \dot{J}_0(t), \tag{37}$$

where $\Phi|_{\sigma=0} = \Phi(t, r, \sigma = 0)$. The action thus gives a corner contribution to the $r$-boundary action (32).

The action (32) depends only on the boundary values of the fields, so from now on when referring to $\Phi$, $X$ and $Y$ we exclusively mean their boundary values. The boundary conditions $a_\sigma = \mathcal{J}(t, \sigma) L_0$ impose the following conditions on these boundary fields

$$\Phi' = \mathcal{J}, \qquad X' = 0, \qquad Y' + Y\Phi' = 0. \tag{38}$$

These constraints remove the second term in the boundary action (32). Including the $\sigma$-boundary term (37), the total action (31) simplifies to

$$S_{\text{red}}[\Phi] = -\frac{k}{4\pi} \int_{\partial \mathcal{M}} dt d\sigma \frac{1}{2} \dot{\Phi} \Phi' + \frac{k}{4} \int dt \, \Phi|_{\sigma=0} \dot{J}_0 + I_{\text{bdy}}. \tag{39}$$

And the field $\Phi$ satisfies a generalized periodicity condition

$$\Phi(t, \sigma + 2\pi) = \Phi(t, \sigma) + 2\pi J_0(t), \tag{40}$$

which captures the holonomy in the bulk. We thus see that the holonomies of the Chern–Simons connection, which correspond to the horizon radii of the BTZ black hole, appear as linear terms in $\sigma$ (or, equivalently, as momenta) in the mode expansion of $\Phi$.

When expressed in terms of a periodic field $\tilde{\Phi}$ defined through $\Phi(t, \sigma) = \tilde{\Phi}(t, \sigma) + \sigma J_0(t)$, the boundary action reads

$$S_{\text{red}}[J_0, \tilde{\Phi}] = -\frac{k}{4\pi} \int_{\Sigma_i} \mathrm{d}t \mathrm{d}\sigma \left( \frac{1}{2} \dot{\tilde{\Phi}}(\tilde{\Phi}' + 2J_0) \right) + I_{\text{bdy}} \tag{41}$$

up to total time derivatives. This action is equivalent to the boundary action obtained from reducing U(1) Chern–Simons theory to the boundary [45].

## 4 Boundary Hamiltonians

In this section we fix the boundary term $I_{\text{bdy}}$ in the Hamiltonian form of the Chern–Simons action (24). The defining property of $I_{\text{bdy}}$ is that its variation cancels the boundary term obtained from the variation of the bulk Chern–Simons theory,

$$\delta I_{\text{bdy}} = \frac{k}{2\pi} \int_{\partial \mathcal{M}} \mathrm{d}t \mathrm{d}\sigma \, \text{tr}(a_t \, \delta a_\sigma). \tag{42}$$

For our near horizon boundary conditions in section 2.3 the only term contributing to the variation of $a_\sigma$ is $\delta \mathcal{J} L_0$ and hence only the $L_0$ component of $a_t$ will contribute to the boundary Hamiltonian. If we write $a_t = -\zeta(t, \varphi) L_0$ the variation of the boundary term becomes

$$\delta I_{\text{bdy}} = -\frac{k}{4\pi} \int_{\partial \mathcal{M}} \mathrm{d}t \mathrm{d}\sigma \, \zeta \, \delta \mathcal{J}. \tag{43}$$

The remaining part of the specification of the boundary conditions consists of stating whether $\zeta$ is allowed to be a functional of $\mathcal{J}$ and if so, which one. A minimal requirement that we impose is finiteness and integrability of the boundary term $I_{\text{bdy}}$. Finiteness of (43) is guaranteed by our choice of radial gauge (13). Integrability imposes a condition on $\zeta$,

$$\zeta(\mathcal{J}) = \frac{\delta \mathcal{H}}{\delta \mathcal{J}}, \tag{44}$$

where $\mathcal{H}$ is the boundary Hamiltonian density that we shall refer to as "near horizon Hamiltonian density" when placing the boundary at or near the horizon.

There are infinitely many different choices of $\zeta(\mathcal{J})$ that lead to an integrable boundary term. We discuss a few of them in this section and give their gravitational interpretation, starting with the choice that is most natural from a near horizon perspective.

### 4.1 Near horizon Hamiltonian

The simplest assumption is $\delta \zeta = 0$, corresponding to the near horizon boundary conditions of [22]; additionally we assume $\zeta = $ constant. The variation of the boundary term is trivially integrated

$$I_{\text{bdy}} = -\int \mathrm{d}t \, H_{\text{NH}} = -\int \mathrm{d}t \mathrm{d}\sigma \, \mathcal{H}_{\text{NH}} = -\frac{k}{4\pi} \int \mathrm{d}t \mathrm{d}\sigma \, \zeta \mathcal{J} = -\frac{k}{2} \int \mathrm{d}t \, \zeta J_0, \tag{45}$$

to obtain the near horizon Hamiltonian

$$H_{\text{NH}} = \frac{k}{2}\zeta J_0 \,. \tag{46}$$

As $J_0$ commutes with all other modes $J_n$ the Hamiltonian (46) assigns equal energy to each of the $J_n$ descendants of a state, which is the softness property mentioned in section 2.3.

In terms of the boundary scalar field theory (39), the near horizon action is given by

$$S_{\text{NH}}[\Phi] = -\frac{k}{4\pi}\int dt d\sigma \,\frac{1}{2}\dot{\Phi}\Phi' - \frac{k}{2}\int dt \left(\zeta J_0 - \frac{1}{2}\Phi|_{\sigma=0}\dot{J}_0\right)\,. \tag{47}$$

Note that the last terms do not contribute to the near horizon equations of motion for $\Phi$, but it fixes the time derivative of $\Phi_0$, the zero mode of $\Phi$.

$$\dot{\Phi}' = 0\,, \qquad \dot{\Phi}_0 = -\zeta\,. \tag{48}$$

The solution to the first field equations (48) is $\Phi(t,\sigma) = \Phi_t(t) + \Phi_\sigma(\sigma)$ which, together with the periodicity condition (40), implies $\dot{J}_0 = 0$ and gives the mode decomposition announced in (3),

$$\Phi(t,\sigma)\Big|_{\text{EOM}} = \Phi_0(t) + J_0\sigma + \sum_{n\neq 0}\frac{J_n}{in}e^{in\sigma}\,, \tag{49}$$

with $\Phi_0(t) = -\zeta t + \phi$ where $\phi$ is an arbitrary constant.

Since the near horizon Hamiltonian density (2) is a total derivative term, the only non-trivial information in our near horizon theory is captured by the on-shell value of the Hamiltonian and by the symplectic structure. To discuss the latter we make off-shell a mode decomposition like (49),

$$\Phi(t,\sigma) = \Phi_0(t) + J_0(t)\sigma + \sum_{n\neq 0}\Phi_n(t)e^{in\sigma}\,, \qquad \Phi_n(t) := \frac{J_n(t)}{in}\,, \tag{50}$$

where we allow arbitrary time-dependence of $J_n$, and plug it into the near horizon action (47), obtaining

$$S_{\text{NH}}[\Phi_0,J_n] = \frac{k}{2}\int dt \left(-\dot{\Phi}_0 J_0 - \sum_{n>0}\dot{\Phi}_n J_{-n} - \zeta J_0\right)\,. \tag{51}$$

The Hamiltonian action corresponding to (51) simplifies to[3]

$$S_{\text{NH}}[\Phi_n,\Pi_n] = \int dt \left(\sum_{n\geq 0}\Pi_n\dot{\Phi}_n - H_{\text{NH}}\right)\,, \tag{52}$$

with the near horizon Hamiltonian (46) and the momenta

$$\Pi_n = -\frac{k}{2}J_{-n} \qquad n\geq 0\,, \tag{53}$$

which are the main results of this paper announced in the introduction (1)-(3). Canonical Poisson brackets $\{\Phi_0,\Pi_0\} = 1$, $\{\Phi_n,\Pi_m\} = \delta_{n,m}$ then essentially recover (one chiral half of) the near horizon symmetry algebra (18).

$$i\{J_n,J_m\} = \frac{2}{k}n\,\delta_{n+m,0}\,, \qquad i\{J_0,\Phi_0\} = \frac{2i}{k}\,. \tag{54}$$

---

[3]The sum extends only over positive integers to avoid having to go through the Dirac analysis of systems with second class constraints, i.e., the configuration variables are defined as positive index quantities $J_n$ and the momenta, up to a scale factor, by negative index quantities $J_{-n}$. This is a trivial implementation of the more general Faddeev–Jackiw method [48].

The only change as compared to the near horizon symmetry algebra (18) is the presence of a canonically conjugate, $\Phi_0$, for the zero mode $J_0$. The relation (53), which basically states that the momentum $\Pi$ of the scalar field $\Phi$ is given by its spatial derivative, is a key characteristic of self-dual scalar fields [47], so the action (52) resembles the Floreanini–Jackiw action [27]. We shall say a bit more about this relation in the next subsection when discussing Brown–Henneaux type of boundary conditions from a near horizon perspective.

For later purposes it is convenient to split the scalar field

$$\Phi = \Phi_+ + \Phi_- , \tag{55}$$

into positive and negative modes (the subscripts $\pm$ should not be confused with the superscripts $\pm$ referring to the chiral sectors, which we mostly suppress in this work)

$$\Phi_+(t, \sigma) = \Phi_0(t) + \sum_{n>0} \frac{J_n(t)}{in} e^{in\sigma} , \qquad \Phi_-(t, \sigma) = \sigma J_0(t) - \sum_{n>0} \frac{J_{-n}(t)}{in} e^{-in\sigma} , \tag{56}$$

and define the momentum

$$\Pi := -\frac{k}{4\pi} \Phi'_- = -\frac{k}{4\pi} \Big( J_0(t) + \sum_{n>0} J_{-n}(t) e^{-in\sigma} \Big), \tag{57}$$

in terms of which the symplectic part of the action (52) is given by $\int dt d\sigma \, \Pi \dot{\Phi}_+$.

Since the Hamiltonian $H_{\text{NH}}$ in (46) commutes with all the oscillators, $J_n$ descendants cannot raise or lower the energy of any state in the theory. We thus recover the expected statement that $J_n$ descendants are soft hair on the black hole horizon.

With possible applications to microstates and black hole entropy in mind this soft hair degeneracy is a stumbling block, since there is no sensible way to count the number of soft states contributing to a black hole at a given energy $J_0$. Proposals to lift the degeneracy of the soft hairs were made in the literature [31, 32]. In section 5 below we propose a new way for lifting the soft hair degeneracy by constructing a natural hierarchy of integrable boundary terms (44), and then taking the limit to the near horizon Hamiltonian above. In order to achieve this we introduce this hierarchy in the remainder of this section, starting with a reconsideration of Brown–Henneaux boundary conditions from a near horizon perspective.

## 4.2 Chiral bosons, Liouville theory and Schwarzian action

To set the stage for a generalization of the boundary action, we discuss the relation of the above reduction to the more familiar reduction of the action using Brown–Henneaux boundary conditions, leading to Liouville theory [26].

In [22] it was shown how the near horizon boundary conditions are related to the usual Brown–Henneaux boundary conditions with chemical potentials. Using slightly different conventions here, we map the near horizon boundary conditions $a_\sigma^{\text{NH}}$ given by (15) to the Brown–Henneaux ones $[a_\sigma^{\text{BH}}$ given in (112)$]$ by a suitable gauge transformation $g$,

$$a_\sigma^{\text{BH}} = g^{-1}(\partial_\sigma + a_\sigma^{\text{NH}})g . \tag{58}$$

This map relates the charges in both formulations of the boundary conditions as

$$\mathcal{L} = \frac{1}{4}\mathcal{J}^2 + \frac{1}{2}\mathcal{J}' . \tag{59}$$

The chemical potentials are also related to each other, but in a state-dependent way; i.e. the relation involves the charges $\mathcal{J}$

$$\zeta = \mu' - \mathcal{J}\mu . \tag{60}$$

Assuming $\delta\mu = 0$ and integrating (44) yields the Brown–Henneaux boundary Hamiltonian

$$I_{\text{bdy}} = \frac{k}{4\pi} \int_{\partial\mathcal{M}} \mathrm{d}t\mathrm{d}\sigma\, \mu\left(\frac{1}{2}\mathcal{J}^2 + \mathcal{J}'\right) = \frac{k}{2\pi} \int_{\partial\mathcal{M}} \mathrm{d}t\mathrm{d}\sigma\, \mu\mathcal{L}. \tag{61}$$

Expressed in terms of the scalar field $\Phi$, one chiral half of the reduced action with Brown–Henneaux boundary conditions and arbitrary chemical potential is given by a boundary action

$$S[\Phi] = \int_{\partial\mathcal{M}} \mathrm{d}t\mathrm{d}\sigma\, \left(\Pi\dot{\Phi}_+ + \frac{\mu k}{8\pi}\left((\Phi')^2 + 2\Phi''\right)\right), \tag{62}$$

analogous to the the near horizon action (47), but with a different Hamiltonian density,

$$\mathcal{H}_{\text{BH}} = -\frac{k\mu}{8\pi}\left((\Phi')^2 + 2\Phi''\right). \tag{63}$$

It is also possible to derive this boundary action directly from imposing the constraints on $a_\sigma$ and $a_t$ following from the Brown–Henneaux boundary conditions (112) and (117). The constraints on the fields $X, Y$ and $\Phi$ appearing in the Gauss decomposition (30) for the Brown–Henneaux boundary conditions are

$$X' = e^{-\Phi}, \qquad \Phi' = -2Y, \tag{64}$$

and $\mathcal{L}$ is related to the field $\Phi$ by the Miura transformation (or, equivalently, a twisted Sugawara construction)

$$\mathcal{L} = \frac{1}{4}\left(\Phi'\right)^2 + \frac{1}{2}\Phi''. \tag{65}$$

Implementing these constraints into the action (28) gives the action (62) up to total derivatives. The periodicity condition (40) on the field $\Phi$ implies

$$X(\sigma + 2\pi) = e^{-2\pi J_0}X(\sigma). \tag{66}$$

Here the holonomy has been parameterized by (34), and $J_0$ is related to the zero modes of $\mathcal{L}_0$ via equation (59).

Under conformal transformations generated by $\xi$ the energy-momentum flux component $\mathcal{L}$ transforms with an infinitesimal Schwarzian derivative, while $X$ transforms like a weight-0 primary, $\Phi$ like a twisted weight-0 primary,[4] $e^{-\Phi}$ like a weight-1 primary and $Y$ like a twisted weight-1 primary. The corresponding transformation laws are given by (115) and

$$\delta_\xi X = \xi X', \qquad\qquad \delta_\xi \Phi = \xi\Phi' - \xi', \tag{67}$$

$$\delta_\xi e^{-\Phi} = \left(\xi e^{-\Phi}\right)', \qquad\qquad \delta_\xi Y = \xi Y' + \xi'Y + \tfrac{1}{2}\xi''. \tag{68}$$

The formulas above can be derived starting from the near horizon transformation law (16), using a relation analogous to (60), namely $\eta = \xi\mathcal{J} - \xi'$, with $\mathcal{J} = \Phi'$.

In the case of constant chemical potentials (and discarding holonomy terms)[5] we may compare with known results in the literature. The last term in the action (62) is a total derivative and the reduced action is equal to the Floreanini–Jackiw action [27] of a chiral boson [47] with propagation speed $\mu$. This is a key difference to near horizon boundary conditions, where the propagation speed tends to zero.

---

[4]Entanglement entropy transforms in the same way, see [49]. For vacuum solutions to the Einstein equations $\Phi$ is essentially equivalent to entanglement entropy and the equality (65) corresponds to saturation of the quantum null energy condition [50].

[5]For a recent analysis including the holonomies and both boundaries of the annulus, see [45]

Setting $\mu = 1/\ell$ and combining the two chiral sectors reproduces the results of the reduction under Brown–Henneaux boundary conditions performed in [51], before the two chiral WZW models are combined into a non-chiral WZW model [52].

$$S[\Phi^\pm] = -\frac{k}{8\pi} \int_{\partial \mathcal{M}} \mathrm{d}t\mathrm{d}\sigma \left( \dot{\Phi}^+ \Phi^{+\prime} - \dot{\Phi}^- \Phi^{-\prime} - \frac{1}{\ell}\Phi^{+\prime}\Phi^{+\prime} - \frac{1}{\ell}\Phi^{-\prime}\Phi^{-\prime} \right). \tag{69}$$

It was shown in [51] that for vanishing holonomies this action is related to the Hamiltonian form of Liouville theory by a series of field redefinitions (see (3.7)-(3.11) of [51] for more details). This connects our work to the result of [26].

Another representation of the action (62) is obtained by changing variables to $X$ using the constraint $X' = e^{-\Phi}$. Then the kinetic term $\dot{\Phi}\Phi'$ is equal to the geometric action of the Virasoro group on its coadjoint orbit [53] first derived by Alekseev and Shatashvili [54]

$$S[X] = \frac{k}{4\pi} \int \mathrm{d}t\mathrm{d}\sigma \left( \frac{\dot{X}''}{X'} - \frac{3}{2}\frac{X''\dot{X}'}{X'^2} - \mu\{X,\sigma\} \right), \tag{70}$$

where the Hamiltonian term is given by the Schwarzian derivative

$$\{X,\sigma\} = \frac{X'''}{X'} - \frac{3}{2}\left( \frac{X''}{X'} \right)^2. \tag{71}$$

To obtain the formulation of the Alekseev–Shatashvili action with non-zero representative on the coadjoint orbit one should make a field redefinition $X = \exp(-J_0 f(t,\sigma))$ such that $f(t,\sigma + 2\pi) = f(t,\sigma) + 2\pi$ reproduces the periodicity conditions (66) for $X$. This gives the total action (for $\mu = 1$)

$$S[f;J_0] = \frac{k}{4\pi} \int \mathrm{d}t\mathrm{d}\sigma \left( \frac{\dot{f}''}{f'} - \frac{3}{2}\frac{f''\dot{f}'}{f'^2} - \{f,\sigma\} - \frac{1}{2}J_0^2 f'(\dot{f} - f') \right). \tag{72}$$

The orbit representative, denoted as $b_0$ in [54] is related to the zero mode charges of the bulk solution as

$$b_0 = \frac{k}{8\pi}J_0^2 = \frac{c}{12\pi}\mathcal{L}_0, \tag{73}$$

where $\mathcal{L}_0$ is the zero mode of $\mathcal{L}$. From this formula it is clear that the exceptional $\mathrm{PSL}(2,\mathbb{R})$ invariant orbit at $b_0 = -\frac{c}{48\pi}$ corresponds to the global $\mathrm{AdS}_3$ solution with $\mathcal{L}_0 = -\frac{1}{4}$, while the BTZ black holes correspond to orbits with $b_0 > 0$. Our near horizon boundary conditions do not include the global $\mathrm{AdS}_3$ ground state unless we analytically continue $J_0$ to imaginary values such that $J_0^2 = -1$.

The relation between this action and the boundary theory of pure $\mathrm{AdS}_3$ gravity with Brown–Henneaux boundary conditions was reported in [55] and expanded upon recently in [56]. It is interesting to note that besides the formulation of the boundary action as the geometric action on the coadjoint orbit of the Virasoro group, it can also be obtained as the geometric action for an affine $\hat{u}(1)$ Kac–Moody group. For affine Lie groups the Kirillov–Kostant orbit method gives the WZW model of the corresponding group [57]. In the case of $\hat{u}(1)$, the symplectic term of the geometric action is the near-horizon action (39).[6] As discussed in [55], suitable Hamiltonians for geometric actions are (invariant tensor products of) Noether charges for global symmetries of the symplectic term. In this context the near-horizon Hamiltonian (46) can be understood as the Noether charge for the shift symmetry

$$\Phi(t,\sigma) \to \Phi(t,\sigma) + \phi(t), \tag{74}$$

where $\phi(t)$ is an arbitrary (but fixed) function of time. The Brown–Henneaux Hamiltonian (63) is the square of this Noether charge (up to a total derivative).

---

[6]To be more precise, the near-horizon action corresponds to the term proportional to the central charge of the $\hat{u}(1)$ geometric action. The orbit representative term can be obtained by a field redefinition to a periodic field $\varphi$ as $\Phi(t,\sigma) = \varphi(t,\sigma) + J_0\sigma$.

## 4.3 KdV action and symmetries

We generalize now the key relation (60) between near horizon and Brown–Henneaux chemical potentials in a specific non-linear way, while maintaining finiteness and integrability of the boundary charges as well as the shift symmetry (74). This will lead to novel boundary actions. The Hamiltonian of the boundary theory is modified by choosing boundary conditions where the chemical potentials $\zeta$ depend on the charges $\mathcal{J}$. The choice (60) in the previous subsection was a special case of the general possibility (44). In [58] (see also [59,60]) similar arguments were used to derive the KdV hierarchy from boundary conditions on AdS$_3$ gravity, where the Brown–Henneaux charges $\mathcal{L}$ solve the KdV equation. In this subsection we show how the boundary action (39) reduces to the action principle leading to the KdV equation for the near horizon charges $\mathcal{J}$ instead of for $\mathcal{L}$.

The idea is to choose the chemical potentials $\zeta$ such that the boundary term (25) integrates to a differential polynomial of rank $N$ representing the integral of motion of the KdV equation. A useful basis of chemical potentials $\zeta$ are the Gelfand–Dikii differential polynomials $R_N(\mathcal{J})$ [61], as they automatically satisfy the integrability condition (44).

Starting from $R_0 = 1$, these differential polynomials are defined recursively by

$$R'_{N+1} = \frac{N+1}{2N+1} \mathcal{D} R_N \,, \tag{75}$$

for $\mathcal{D} = \partial_\sigma \mathcal{J} + 2\mathcal{J}\partial_\sigma + \frac{1}{2}\partial_\sigma^3$. Taking

$$\zeta_N = R_N(\mathcal{J}), \tag{76}$$

and integrating (44) leads to a hierarchy of boundary Hamiltonians

$$H_0 = \int \mathrm{d}\sigma\, \mathcal{J}, \tag{77}$$

$$H_1 = \int \mathrm{d}\sigma\, \frac{1}{2}\,\mathcal{J}^2, \tag{78}$$

$$H_2 = \int \mathrm{d}\sigma \left(\frac{1}{3}\,\mathcal{J}^3 - \frac{1}{6}\,\mathcal{J}'^2\right), \tag{79}$$

$$H_N = \frac{1}{N+1} \int \mathrm{d}\sigma\, R_{N+1}(\mathcal{J}). \tag{80}$$

The Hamiltonian $H_0$ is identical to the near horizon Hamiltonian (46) for constant $\zeta$. The Hamiltonian $H_1$ (when multiplied by constant $\mu$) leads to the chiral boson action (62) following from the Brown–Henneaux boundary conditions, which are now understood as deformed boundary conditions on the horizon. The Hamiltonian $H_2$ [after rescaling by $-k/(4\pi)$ and using the definitions (55)-(57)] leads to the boundary action

$$S_{\mathrm{KdV}}[\Phi] = \int \mathrm{d}t\mathrm{d}\sigma \left(\Pi\,\dot{\Phi}_+ + \frac{k}{4\pi}\left(\frac{1}{3}\left(\Phi'\right)^3 - \frac{1}{6}\left(\Phi''\right)^2\right)\right). \tag{81}$$

The field equations for $\Phi$ following from the KdV action (81) can be written in terms of $\mathcal{J} = \Phi'$ as

$$\dot{\mathcal{J}} = 2\mathcal{J}\mathcal{J}' + \frac{1}{3}\,\mathcal{J}'''. \tag{82}$$

This is the KdV equation for the current $\mathcal{J}$. In this case the chemical potential is $\zeta_2 = \mathcal{J}^2 + \frac{1}{3}\mathcal{J}''$ which means the bulk field equations $F_{t\varphi} = 0$ also reproduce the KdV equation (82). Finally, the identity (80) was derived in [61], see their Eq. (16') [taking into account the respective

normalizations of $R_N$] and their appendix 1. Appendix B contains a discussion of Gelfand–Dikii differential polynomials and associated Hamiltonian densities for general $N$.

Thus, we have constructed a (KdV) hierarchy of different boundary conditions labelled by a non-negative integer $N$, with the case $N = 0$ corresponding to near horizon boundary conditions, $N = 1$ to Brown–Henneaux boundary conditions, $N = 2$ to KdV boundary conditions and $N > 2$ leading to boundary Hamiltonian densities of the form

$$\mathcal{H}_N[\Phi] \sim \frac{1}{N+1}\mathcal{J}^{N+1} + \sum_{i=1}^{N-1} h_{i,N}\mathcal{J}^{N-i-1}\left(\partial_\sigma^i \mathcal{J}\right)^2 + \mathcal{H}_N^{\text{nl}}, \qquad \mathcal{J} = \Phi' \qquad (83)$$

with some rational coefficients $h_{i,N}$ and additional terms of similar form in $\mathcal{H}_N^{\text{nl}}$ when $N \geq 5$, see appendix B for details. The equations of motion

$$\dot{\mathcal{J}} = N\,\mathcal{J}^{N-1}\mathcal{J}' + \sum_{i=1}^{N-2} \hat{h}_{i,N}\mathcal{J}^{N-i-1} \overset{\leftrightarrow}{\partial}_\sigma^{2i+1} \mathcal{J} + \frac{2(N!)^2}{(2N)!}\partial_\sigma^{2N-1}\mathcal{J}, \qquad (84)$$

descending from the action

$$S_N[\Phi] = -\frac{k}{4\pi}\int \mathrm{d}t\mathrm{d}\sigma\left(\frac{1}{2}\dot{\Phi}\Phi' - \mathcal{H}_N[\Phi]\right), \qquad (85)$$

generalize the KdV equation (82), where $\hat{h}_{i,N}$ are rational multi-coefficients and $\overset{\leftrightarrow}{\partial}$ means that some derivatives may act to the left. The expression on the right hand side of the equations of motion (84) is given by $\partial_\sigma R_N$, see appendix B for explicit results up to $N = 6$.

The action (85) (up to a total derivative term) again has the shift symmetry (74). The field equations (84) for $N > 1$ have an anisotropic scale invariance with odd anisotropy coefficient

$$N > 1: \qquad t \to \lambda^{2N-1}\,t, \qquad \sigma \to \lambda\sigma, \qquad \Phi \to \lambda^{-1}\Phi. \qquad (86)$$

The Lifshitz type scaling behavior (86) resembles the one found in [58], but differs from it since our basic entity is the spin-1 current $\mathcal{J} = \Phi'$, whereas in [58] the basic entity was the spin-2 current $\mathcal{L}$. Note that (86) is an invariance of the equations of motion, but not of the action (85), which gets multiplied by a factor $1/\lambda^2$ (see [62] for a discussion of such invariances). For $N = 1$ the scale invariance becomes isotropic, and the transformation weight of $\mathcal{J}$ could be arbitrary. For $N = 0$ additionally the transformation weight of time becomes arbitrary. To fix this arbitrariness, for $N = 0$ and $N = 1$ we demand that not only the equations of motion are invariant, but also the action (85), obtaining

$$N = 0 \text{ or } 1: \qquad t \to \lambda^N\,t, \qquad \sigma \to \lambda\sigma, \qquad \Phi \to \Phi. \qquad (87)$$

We consider finally the near horizon symmetries induced by deformed boundary conditions within the KdV hierarchy, starting with the two known cases. For $N = 0$ the near horizon boundary conditions remain undeformed and the near horizon symmetries are given by spin-1 currents (18). For $N = 1$ the near horizon boundary conditions are deformed to Brown–Henneaux and the near horizon symmetries are given by spin-2 currents

$$L_n = \frac{k}{4}\sum_p J_{n-p}J_p + \dots, \qquad (88)$$

where the ellipsis denotes a possible twist term proportional to $nJ_n$. The Sugawara relation (88) is compatible with the Miura-transformation (65) and with the boundary Hamiltonian $H_1$ (78), and leads to the Poisson brackets

$$i\{L_n, L_m\} = (n-m)L_{n+m} + \dots, \qquad (89)$$

$$i\{L_n, J_m\} = -mJ_{n+m} + \dots, \qquad (90)$$

where the ellipses denote possible central extensions depending on the twist term. Since (89) is nothing but the Virasoro algebra, $L_n$ are spin-2 currents, with their usual brackets with the spin-1 current (90).

For $N = 2$ the situation is qualitatively different from $N = 0$ and $N = 1$, since one obtains infinitely many mutually commuting charges, namely all the $H_i$ [61]. Note that, for instance, $H_0$ is the zero mode of the tower of $N = 0$ charges $J_n$, $H_1$ is the zero mode of the tower of $N = 1$ charges $L_n$, and $H_i$ (83) for $i \geq 2$ are zero mode charges given by the whole hierarchy of boundary Hamiltonians. For $N > 2$ the situation is identical to $N = 2$. Similar results were derived for a KdV hierarchy based on the Brown–Henneaux boundary charges by Perez, Tempo and Troncoso in [58], where our $\hat{u}(1)$ charges $\mathcal{J}$ are replaced by the Virasoro charges $\mathcal{L}$. Most of their conclusions carry over to the present case, including the statements made in this paragraph.

# 5 KdV scaling limit for the near horizon Hamiltonian

In this section we consider analytic continuation of the family of boundary Hamiltonian densities (83) to real $N \in [0, 1]$ with the intention of taking the limit $N = \epsilon \to 0^+$, assuming large $\mathcal{J}$. We start by omitting all derivative terms in $\mathcal{J}$. There are three reasons for this.

1. For the boundary points of the interval of interest, $N = 0, 1$, the derivative terms are absent, so it does make sense to assume also the analytic continuation between these two points maintains this property.

2. In the limit of large black holes (which is necessary for a good semi-classical description) the quantity $\mathcal{J}$ parametrically is large, which means that the first term in (83) dominates over all the remaining terms.

3. When continuing analytically a good guiding principle is to maintain as many symmetries as possible. The scaling symmetry (87) of the boundary action persists if there are no derivative terms present.

In the continuous family of boundary Hamiltonians

$$H_\epsilon = \frac{k}{4\pi} \frac{\zeta_\epsilon}{\epsilon(1+\epsilon)} \int \mathrm{d}\sigma \, \mathcal{J}^{1+\epsilon}, \qquad 0 \leq \epsilon \leq 1, \quad \zeta_\epsilon \in \mathbb{R}, \tag{91}$$

a convenient normalization factor in front of the integral is introduced in order to have an interesting limit $\epsilon \to 0^+$,

$$\lim_{\epsilon \to 0^+} H_\epsilon = \frac{k}{4\pi} \zeta_\epsilon \int \mathrm{d}\sigma \, \mathcal{J} \ln \mathcal{J} =: H_{\log}, \tag{92}$$

where we dropped a boundary term before taking the limit. The associated action at finite $\epsilon$ [using again the definitions (55)-(57)]

$$S_\epsilon[\Phi] = \int \mathrm{d}t \mathrm{d}\sigma \left( \Pi \dot{\Phi}_+ - \frac{k}{4\pi} \frac{\zeta_\epsilon}{\epsilon(1+\epsilon)} (\Phi')^{1+\epsilon} \right), \tag{93}$$

yields the limiting action

$$\lim_{\epsilon \to 0^+} S_\epsilon[\Phi] = \int \mathrm{d}t \mathrm{d}\sigma \left( \Pi \dot{\Phi}_+ - \frac{k}{4\pi} \zeta_\epsilon \Phi' \ln(\Phi') \right) =: S_{\log}[\Phi]. \tag{94}$$

The action (93) is invariant under anisotropic scalings

$$t \to \lambda^\epsilon t, \qquad \sigma \to \lambda \sigma, \qquad \Phi \to \Phi. \tag{95}$$

For $\epsilon = 1, 0$ the result (87) is recovered. Also the limiting action (94) has this invariance for constant $\zeta_\epsilon$, since the inhomogeneous term coming from the logarithm is a total derivative term. Gratifyingly, in the limit of vanishing $\epsilon$ the scale invariance above is compatible with the one of near horizon boundary conditions [40, 58]

The field equations derived from the limiting action (94),

$$\dot{\Phi}' = -\zeta_\epsilon \frac{\Phi''}{\Phi'}, \tag{96}$$

can easily be integrated once,

$$\dot{\Phi} = -\zeta_\epsilon (1 + \ln \Phi'), \tag{97}$$

where the integration constant is fixed by the $J_0$ field equation.

The off-shell Fourier-like expansion (50) for $\Phi$ permits to decompose the integrated field equations (97) in the limit of large $J_0$,

$$\dot{\Phi}_0 = -\zeta_\epsilon (1 + \ln J_0), \tag{98a}$$

$$\dot{J}_0 = 0, \tag{98b}$$

$$\dot{J}_n = -in\,\zeta_\epsilon \frac{J_n}{J_0} + \dots, \tag{98c}$$

with the solution

$$\Phi(t, \sigma) = \Phi_0(t) + J_0(t)\sigma + \sum_{n\neq0} \frac{J_n(t)}{in} e^{in\sigma} \overset{\text{EOM}}{=} \phi(t) + J_0\sigma + \sum_{n\neq0} \frac{J_n^{(0)}}{in} e^{in\left(\sigma - \zeta_\epsilon t/J_0\right)} + \dots, \tag{99}$$

where $\phi(t) = -\zeta_\epsilon (1 + \ln J_0) t + \phi_0$, the ellipses denote subleading terms in $1/J_0$, and $J_n^{(0)}$ are integration constants of individual soft hair Fourier modes. Note that $\dot{J}_0$ vanishes as consequence of the field equations, so it is not an assumption but rather a result that the quantity determining the holonomy at $\epsilon = 0$ is time independent.

The main physical consequence of the limiting action (94) as compared to the near horizon action (52) is that the soft hair excitations acquire a positive energy, which we calculate now.

Plugging the mode expansion (50) into the Hamiltonian (92) yields

$$H_{\text{log}} = \frac{k\zeta_\epsilon}{4\pi} \int d\sigma \left( J_0(t) + \sum_{n\neq0} J_n(t) e^{in\sigma} \right) \left( \sum_{n\neq0} \frac{J_n(t)e^{in\sigma}}{J_0(t)} - \frac{1}{2} \left( \sum_{n\neq0} \frac{J_n(t)e^{in\sigma}}{J_0(t)} \right)^2 + \dots \right), \tag{100}$$

where the ellipsis refers to higher order terms suppressed at least by $1/J_0^3$ and to terms that exclusively depend on the zero mode $J_0$ and thus do not contribute to the dynamics of soft hair excitations. Neglecting these terms, the Hamiltonian (100) integrates to

$$H_{\text{log}} = \frac{k\zeta_\epsilon}{2J_0(t)} \sum_{n>0} J_n(t) J_{-n}(t) \overset{\text{EOM}}{=} \frac{k\zeta_\epsilon}{2J_0} \sum_{n>0} J_n^{(0)} J_{-n}^{(0)}. \tag{101}$$

Up to an overall factor, the term displayed in (101) is essentially the Sugawara stress tensor of a spin-1 current. If expressed in terms of momenta (53) the limiting Hamiltonian

$$H_{\text{log}}[J_n, \Pi_n] = \frac{ik\zeta_\epsilon}{4\Pi_0} \sum_{n>0} n J_n \Pi_n \tag{102}$$

is of '$xp$-form'. The limiting Hamiltonian action

$$S_{\log}[\Phi_n, \Pi_n] = \int dt \Big( \sum_{n \geq 0} \dot{\Pi}_n J_n - H_{\log} \Big), \tag{103}$$

recovers the on-shell condition $J_0 = $ const. and produces equations of motion for the soft hair excitations

$$\dot{J}_n = -in\zeta_\epsilon \frac{J_n}{J_0}, \qquad \qquad \dot{\Pi}_n = in\zeta_\epsilon \frac{\Pi_n}{J_0}, \tag{104}$$

that coincide with (98), up to terms suppressed by $1/J_0^2$.

Thus, soft hair excitations of the vacuum, $J_{-n}|0\rangle$, are not soft with respect to $H_{\log}$, but rather are finite energy eigenstates,[7]

$$H_{\log} J_{-n}|0\rangle = [H_{\log}, J_{-n}]|0\rangle = \frac{\zeta_\epsilon}{J_0} n J_{-n}|0\rangle, \tag{105}$$

with eigenvalues proportional to $n$.

# 6 Discussion

We conclude with a comparison and intriguing relations to previous results and proposals, in particular the fluff proposal [31,32], starting with the latter.

The fluff proposal envisions the BTZ microstates as $\hat{u}(1)$ descendents of the vacuum

$$\big|\text{BTZ micro}(\{n_i^\pm\})\big\rangle = \prod J_{-n_i^+}^+ J_{-n_i^-}^- |0\rangle, \qquad J_n^\pm |0\rangle = 0, \quad \forall n \geq 0, \tag{106}$$

labelled by two sets of positive integers $\{n_i^\pm\}$ subject to the spectral constraints

$$\sum n_i^\pm = c\,\Delta^\pm, \qquad \Delta^\pm = \frac{1}{2}\big(\ell M_{\text{BTZ}} \pm J_{\text{BTZ}}\big) = \frac{c}{24}\big(J_0^\pm\big)^2, \tag{107}$$

that provide a cutoff on these descendents. (It is assumed that the products of Brown–Henneaux central charge $c$ and weights $\Delta^\pm$ are large integers; $M_{\text{BTZ}}$ and $J_{\text{BTZ}}$ are Brown–Henneaux mass and angular momentum.) The spectral constraints (107) give soft hair excitations an effective energy linear in the mode number $n$. This property leads to a Hardy–Ramanujan counting [63] of the degeneracy of $\hat{u}(1)$ descendents (dubbed "fluff") leading to the Bekenstein–Hawking entropy of BTZ black holes. While in [31] the constraints (107) were imposed essentially by an argument going back to Bañados [64], in [32] they were derived from independent working assumptions. One of them required the existence of a weight-1 CFT primary $\mathcal{W} = \exp(-\Phi)$ with the twisted periodicity property $\mathcal{W}(\tau, \sigma+2\pi) = \exp(-2\pi J_0)\,\mathcal{W}(\tau, \sigma)$ and non-vanishing commutation relation between the zero mode operator $\hat{\Phi}_0$ of $\Phi$ and the zero mode operator $\hat{J}_0$ of the $\hat{u}(1)$ current.

Key aspects of the fluff proposal reproduced by our limiting action (103) with (102) are

1. the fact that soft hair excitations fall into $\hat{u}(1)$ current algebra representations (54)

2. the existence of a canonically conjugate to the near horizon zero mode charge $J_0$, namely $\Phi_0$, with Poisson bracket (54)

3. the existence of a weight-1 CFT primary operator $X' = \mathcal{W} = \exp(-\Phi)$ in (64) with generalized periodicity property (40)

---

[7]In evaluating the commutator (105) we made the canonical replacement $i\{,\} \to [,]$ and used the left Poisson bracket (54), since we still have the same symplectic structure as in the undeformed theory.

4. the lift of soft hair degeneracy to energies linear in the mode number $n$, see (105).

Additionally, the anisotropic scale invariance of the original near horizon action (47) is shared by the limiting action (94), which played a crucial rôle in a Cardy-type of counting of BTZ microstates from a near horizon perspective [65]. We have thus confirmed crucial aspects of the fluff proposal [31, 32].

However, we have not succeeded in deriving all important aspects of the fluff proposal. In particular, the free parameter $\zeta_\epsilon$ appearing as prefactor in the Hamiltonian (102) is undetermined. In order to obtain the spectral constraints (107) we need to fix it to the value

$$\text{desired:} \quad \zeta_\epsilon^\pm = \frac{J_0^\pm}{c} \tag{108}$$

in each chiral sector, assuming $H_{\log}^\pm \big| \text{BTZ micro}(\{n_i^\pm\}) \big\rangle = \Delta^\pm \big| \text{BTZ micro}(\{n_i^\pm\}) \big\rangle$ for BTZ microstates (106). There are two reasons why the choice (108) is not obvious. First of all, if we keep $J_0$ as state-dependent parameter then it is impossible to demand (108) without deforming the boundary conditions, since $\zeta_\epsilon$ is a chemical potential. Second, even if we allow for possible dependence of $J_0$ the choice (108) is not the most natural one; instead, the commutation relation (105) and the solution (99) both may suggest $\zeta_\epsilon = J_0$ as 'natural'.

We explain now how these issues could be resolved. The first issue is reminiscent of the dilaton gravity description [66] of the SYK model [67–70], the key issue being that the dilaton is allowed to fluctuate, while its zero mode is kept fixed [71, 72]. To resolve this issue, we can simply impose the additional restriction that $J_0$ is kept fixed (like in the microcanonical ensemble) while all other Fourier excitations $J_n$ are allowed to vary. The asymptotic symmetry algebra (18) is compatible with this restriction.

The second issue was already encountered in Carlip's attempt to account for the BTZ black hole entropy, see [73] and refs. therein. Without further input, a $\hat{u}(1)$ current naturally leads to a Virasoro algebra with (quantum) central charge $c = 1$ rather than a (classical) central charge with Brown–Henneaux value $c = 3\ell/(2G)$, which would lead to a considerable under-counting of the degeneracy of microstates.

Therefore, for the fluff proposal to work it is important not only to provide a controlled cutoff on the soft hair spectrum (we have succeeded in doing so in the present work), but also to produce the desired result (108). In [32] this issue was resolved by a set of Bohr-type quantization conditions that led to a counting of a discrete set of conical defect geometries with certain rational values for the conical defect as building blocks for the BTZ microstates. It would be desirable to derive these conditions [or to directly derive (108)] from first principles.

In our way of providing a cutoff for soft hair excitations we used the KdV hierarchy and the gravity-approximation. Since the latter corresponds to the large central charge approximation on the CFT side, it could be rewarding to compare our results with corresponding large $c$ results, such as [74, 75].

Finally, let us point out two different ways of interpreting our near horizon boundary conditions as ultrarelativistic limit of some other theory.

The first one starts from the Floreanini–Jackiw action (62), where the parameter $\mu$ has the physical interpretation as propagation speed of the chiral boson. The ultrarelativistic (or Carrollian) limit is $\mu \to 0$ and recovers our near horizon action (47). This concurs, at least in spirit, with the near horizon analysis by Donnay and Marteau [76] and Penna [77], who found Carrollian structures at black hole horizons.

The second consideration starts from bosonic string theory, where the string spectrum is given by

$$X_\pm^\mu(t \pm \sigma) = \frac{x^\mu}{2} + \frac{\ell_s^2}{2} p_\pm^\mu(t \pm \sigma) + \frac{\ell_s}{\sqrt{2}} \sum_{n \neq 0} \frac{\alpha_{-n}^\pm}{in} e^{in(t \pm \sigma)}. \tag{109}$$

In the naive ultrarelativistic limit $t \to \epsilon t$, $\sigma \to \sigma$, $\epsilon \to 0$ (109) reduces to

$$X_\pm^\mu(\sigma) = \frac{x^\mu}{2} \pm \frac{\ell_s^2}{2} p_\pm^\mu \sigma + \frac{\ell_s}{\sqrt{2}} \sum_{n \neq 0} \frac{\alpha_{-n}^\pm}{in} e^{\pm in\sigma}, \tag{110}$$

which is equivalent to the on-shell expansion (49) for constant $\Phi_0$, identifying $x^\mu = 2\Phi_0$, $\ell_s^2 p_+^\mu = 2J_0$ and $\ell_s \alpha_{-n}^+ = \sqrt{2}J_n$ (and similarly for the other chiral sector). A more careful ultrarelativistic limit also features the linear term in time present in (49) and is relevant for tensionless strings [28], which suggests [78–81] that (nearly) tensionless strings could play a key rôle in the near horizon descriptions of generic black holes and in understanding their microstates.

## Acknowledgements

We are grateful to Hamid Afshar, Martin Ammon, Stephane Detournay, Hernán González, Philip Hacker, Zahra Mirzaiyan, Alfredo Perez, Stefan Prohazka, Max Riegler, Shahin Sheikh-Jabbari, David Tempo, Ricardo Troncoso, Raphaela Wutte, Hossein Yavartanoo and Céline Zwikel for collaboration on near horizon boundary conditions. We thank Arjun Bagchi, Glenn Barnich, Laura Donnay, Oscar Fuentealba, Gaston Giribet, Hernán González, Marc Henneaux, Javier Matulich, Blaja Oblak, Malcolm Perry and Andy Strominger for discussions. We thank particularly one of our referees, Kristan Jensen, for insisting that making the holonomies time-dependent could (and actually did) affect some of our results.

**Funding information**   This work was supported by the FWF projects P30822 and P28751. WM is supported by the ERC Advanced Grant *High-Spin-Grav* and by FNRS-Belgium (convention FRFC PDR T.1025.14 and convention IISN 4.4503.15). Part of this work was completed during the Programme and Workshop "Higher spins and holography" in March/April 2019 at the Erwin Schrödinger International Institute for Mathematics and Physics (ESI), and we thank ESI for hosting us.

## A   Brown–Henneaux boundary conditions and holonomies

As discusssed in the main text, boundary conditions are provided by specifying the form of $a_\sigma$. Being a flat connection, $a_\sigma$ can locally always be written as $g^{-1}\partial_\sigma g$. What distinguishes solutions are the global charges, which in turn are measured by Wilson loops, or holonomies of the connection around a closed $\sigma$-cycle.

$$H_\sigma = \mathrm{tr}\left(\mathcal{P}\exp\oint a_\sigma \, d\sigma\right). \tag{111}$$

Because $a_\sigma$ is an element of the $\mathfrak{sl}(2,\mathbb{R})$ algebra, the holonomy around the $\sigma$-cycle is characterized by the conjugacy classes of $\mathrm{SL}(2,\mathbb{R})$. These are

**Hyperbolic.**   Conjugate to dilatation $g \sim \begin{pmatrix} e^{2\pi\lambda} & 0 \\ 0 & e^{-2\pi\lambda} \end{pmatrix}$, with holonomy $H_\sigma = 2\cosh(2\pi\lambda)$

**Parabolic.**   Conjugate to translation $g \sim \begin{pmatrix} 1 & 2\pi a \\ 0 & 1 \end{pmatrix}$, with holonomy $H_\sigma = 2$

**Elliptic.**  Conjugate to rotation $g \sim \begin{pmatrix} \cos 2\pi\alpha & \sin 2\pi\alpha \\ -\sin 2\pi\alpha & \cos 2\pi\alpha \end{pmatrix}$, with holonomy $H_\sigma = 2\cos(2\pi\alpha)$

To obtain the boundary conditions of Brown and Henneaux we take $a_\sigma$ to be in the 'highest weight gauge' (and similarly for $\bar{a}_\sigma$)

$$a_\sigma = L_+ - \mathcal{L}(t,\sigma)L_- = \begin{pmatrix} 0 & \mathcal{L} \\ 1 & 0 \end{pmatrix}. \tag{112}$$

This choice indeed leads to an asymptotic Virasoro symmetry algebra with the Brown–Henneaux central charge, which can be shown by computing the charges and the variations of $a_\sigma$ under boundary condition preserving gauge transformations $\varepsilon$.

$$\delta_\varepsilon a_\sigma = \partial_\sigma \varepsilon + [a_\sigma, \varepsilon] = \begin{pmatrix} 0 & \delta_\varepsilon \mathcal{L} \\ 0 & 0 \end{pmatrix}. \tag{113}$$

Expanding $\varepsilon = \varepsilon^n L_n$ yields $\varepsilon^+ = \varepsilon^+(\varepsilon^+)$ and

$$\varepsilon^0 = -\partial_\sigma \varepsilon^+, \qquad\qquad \varepsilon^- = \frac{1}{2}\partial_\sigma^2 \varepsilon^+ - \mathcal{L}\varepsilon^+, \tag{114}$$

$$\delta_\varepsilon \mathcal{L} = \partial_\sigma \mathcal{L}\varepsilon^+ + 2\mathcal{L}\partial_\sigma \varepsilon^+ - \frac{1}{2}\partial_\sigma^3 \varepsilon^+. \tag{115}$$

Using this result in the expression for the charges (14) and assuming $\varepsilon^+$ is state-independent, i.e., $\delta\varepsilon^+ = 0$, one may trivially integrate them and finds

$$Q(\varepsilon) = \frac{k}{2\pi}\oint d\sigma\, \varepsilon^+ \mathcal{L}. \tag{116}$$

After expanding the charges in Fourier modes and using the fact that the Poisson brackets of the charges can be computed by $\{Q(\varepsilon_1), Q(\varepsilon_2)\} = \delta_{\varepsilon_1} Q(\varepsilon_2)$, one can easily see the appearance of the Virasoro algebra with a central charge $c = 6k = \frac{3\ell}{2G}$ from the transformation law (115).

The time component part of the connection can be taken to have the same structure as the gauge parameter $\varepsilon$, but with an arbitrary function $\mu$ instead of $\varepsilon^+$. Explicitly,

$$a_t = \varepsilon(\mu) = \mu L_+ - \partial_\sigma \mu L_0 + \left(\frac{1}{2}\partial^2\mu - \mathcal{L}\mu\right)L_-. \tag{117}$$

The metrics (with $\mu = 1$) to which these solutions correspond are the so-called Bañados-metrics [82]. They are obtained by taking $b(r) = \exp(r/\ell\, L_0)$ (and similarly for the barred sector) and plugging the connection into (12)

$$ds^2 = dr^2 - \ell^2 \left(e^{r/\ell}dx^+ - e^{-r/\ell}\bar{\mathcal{L}}(x^-)dx^-\right)\left(e^{r/\ell}dx^- - e^{-r/\ell}\mathcal{L}(x^+)dx^+\right). \tag{118}$$

Here we used light cone coordinates $x^\pm = t/\ell \pm \sigma$.

We can characterize the Bañados solutions by considering the holonomies of the connections $a_\sigma$ and $\bar{a}_\sigma$. The holonomy of the connection (112) (for constant $\mathcal{L}$) is given by

$$H_\sigma(a_\sigma) = \mathrm{tr}\left(\exp(2\pi\sqrt{4\mathcal{L}})L_0\right) = e^{2\pi\sqrt{\mathcal{L}}} + e^{-2\pi\sqrt{\mathcal{L}}}, \tag{119}$$

and likewise for the other sector. This implies that for $\mathcal{L} > 0$ the holonomy falls into the hyperbolic conjugacy class of $\mathrm{SL}(2,\mathbb{R})$. When $\mathcal{L} = 0$, the holonomy is parabolic and for $\mathcal{L} < 0$ it is elliptic.

The Bañados solutions for

$$\mathcal{L} = \frac{2G}{\ell}(\ell\mathfrak{m} + \mathfrak{j}), \qquad \bar{\mathcal{L}} = \frac{2G}{\ell}(\ell\mathfrak{m} - \mathfrak{j}), \qquad (120)$$

correspond to BTZ black holes with mass $\mathfrak{m}$ and angular momentum $\mathfrak{j}$. Thus, generic BTZ black holes have hyperbolic holonomies, whereas extremal black holes (where either $\mathcal{L}$ of $\bar{\mathcal{L}} = 0$) have one connection with parabolic holonomy while the other is still hyperbolic. Whenever $\mathcal{L}$ or $\bar{\mathcal{L}}$ is negative, then the holonomy is conjugate to a complex $SL(2,\mathbb{R})$ element unless $\mathcal{L} = -\frac{1}{4}n^2$ and likewise for $\bar{\mathcal{L}}$. At these points the bulk solution has an angular periodicity of $2\pi n$, with $n = 1$ corresponding to the global $AdS_3$ solution. Any other negative value of $\mathcal{L}, \bar{\mathcal{L}}$ will give conical singularities.

## B   Gelfand–Dikii differential polynomials and Hamiltonians

We list here the first couple of Gelfand–Dikii differential polynomials generated recursively through the defining relation (75)[8]

$$R_0 = 1, \qquad R_1 = \mathcal{J}, \qquad R_2 = \mathcal{J}^2 + \tfrac{1}{3}\mathcal{J}'', \qquad (121)$$

$$R_3 = \mathcal{J}^3 - \tfrac{1}{2}\mathcal{J}'^2 + \left(\mathcal{J}\mathcal{J}'\right)' + \tfrac{1}{10}\mathcal{J}^{(4)}, \qquad (122)$$

$$R_4 = \mathcal{J}^4 - 2\mathcal{J}\mathcal{J}'^2 + \tfrac{1}{5}\mathcal{J}''^2 + \tfrac{2}{3}\left(\mathcal{J}^3\right)'' + \tfrac{2}{5}\left(\mathcal{J}\mathcal{J}''\right)'' + \tfrac{1}{35}\mathcal{J}^{(6)}, \qquad (123)$$

$$R_5 = \mathcal{J}^5 - 5\mathcal{J}^2\mathcal{J}'^2 + \mathcal{J}\mathcal{J}''^2 - \tfrac{1}{14}\mathcal{J}'''^2 + \tfrac{5}{6}\left(\mathcal{J}^4\right)'' + \left(\mathcal{J}^2\mathcal{J}''\right)'' + \tfrac{5}{9}\left(\mathcal{J}'^3\right)'$$
$$+ \tfrac{1}{7}\left(\mathcal{J}\mathcal{J}^{(4)}\right)'' + \tfrac{1}{7}\left(\mathcal{J}'\mathcal{J}^{(4)}\right)' + \tfrac{13}{42}\left(\mathcal{J}''^2\right)'' + \tfrac{1}{126}\mathcal{J}^{(8)}, \qquad (124)$$

$$R_6 = \mathcal{J}^6 - 10\mathcal{J}^3\mathcal{J}'^2 + 3\mathcal{J}^2\mathcal{J}''^2 - \tfrac{3}{7}\mathcal{J}\mathcal{J}'''^2 + \tfrac{1}{42}\left(\mathcal{J}^{(4)}\right)^2 - \tfrac{5}{6}\mathcal{J}'^4 + \tfrac{30}{63}\mathcal{J}''^3$$
$$+ \left(\mathcal{J}^5\right)'' + 2\left(\mathcal{J}^3\mathcal{J}''\right)'' + \tfrac{10}{3}\left(\mathcal{J}\mathcal{J}'^3\right)' + \tfrac{1}{7}\left(3\mathcal{J}^2\mathcal{J}^{(4)} + 6\mathcal{J}\mathcal{J}'\mathcal{J}''' + 7\mathcal{J}'^2\mathcal{J}''\right)''$$
$$+ \tfrac{1}{21}\left(60\mathcal{J}\mathcal{J}''\mathcal{J}''' + 9\mathcal{J}'\mathcal{J}''^2 + \mathcal{J}\mathcal{J}^{(7)} + 3\mathcal{J}'\mathcal{J}^{(6)} + 8\mathcal{J}''\mathcal{J}^{(5)} + 11\mathcal{J}'''\mathcal{J}^{(4)}\right)'$$
$$+ \tfrac{1}{462}\mathcal{J}^{(10)}, \qquad (125)$$

$$\vdots = \vdots$$

$$R_N = \mathcal{J}^N + N\sum_{i=1}^{N-2}(-1)^i\frac{((i+1)!)^2}{(2i+2)!}\binom{N-1}{i+1}\mathcal{J}^{N-i-2}\left(\partial_\sigma^i\mathcal{J}\right)^2 + N\,\mathcal{H}_{N-1}^{\mathrm{nl}} + \partial_\sigma R_N^{\mathrm{td}}, \qquad (126)$$

where $\mathcal{H}_{N-1}^{\mathrm{nl}}$ is defined below and $R_N^{\mathrm{td}}$ captures total derivative terms. Their schematic form for $N > 2$ is given by

$$R_N^{\mathrm{td}} = \sum_{\{b_{i,N}\}} r_{\{b_{i,N}\}}\,\mathcal{J}^{b_{0,N}}\prod_{i=1}^{2N-5}\left(\partial_\sigma^i\mathcal{J}\right)^{b_{i,N}} + \frac{2^{1-N}N!}{(2N-1)!!}\partial_\sigma^{2N-3}\mathcal{J}, \qquad (127)$$

where the sets of non-negative integers $\{b_{i,N}\}$ are subject to the constraints

$$\sum_{i=0}^{2N-5}(i+2)\,b_{i,N} = 2N-1, \qquad (128)$$

and $r_{\{b_{i,N}\}}$ are rational coefficients for each of these sets. The expression $\mathcal{H}_N^{\mathrm{nl}}$ denotes terms at least cubic in derivatives of $\mathcal{J}$. These terms vanish for $N < 5$ and otherwise read

$$\mathcal{H}_N^{\mathrm{nl}} = \sum_{\{a_{i,N}\}} h_{\{a_{i,N}\}}\,\mathcal{J}^{a_{0,N}}\prod_{i=1}^{N-4}\left(\partial_\sigma^i\mathcal{J}\right)^{a_{i,N}} + h_{\{a_{2,N}=1,\,a_{N-3,N}=2\}}\,\mathcal{J}''\left(\partial_\sigma^{N-3}\mathcal{J}\right)^2, \qquad (129)$$

---

[8]The coefficient of the third-derivative term in $\mathcal{D}$ defined below (75) is free. We fixed it to $\frac{1}{2}$, which, together with our normalization choice for $R_N$, explains differences to (and between) results in the literature.

where the sets of non-negative integers $\{a_{i,N}\}$ are subject to the constraints

$$\sum_{i=0}^{N-4}(i+2)\,a_{i,N} = 2N+2\,, \qquad \sum_{i=1}^{N-4} a_{i,N} \geq 3\,, \qquad a_{i^{\max},N} \geq 2\,, \tag{130}$$

with $a_{i^{\max},N}$ denoting the number for the largest value of $i = i^{\max}$ that leads to non-vanishing $a_{i,N}$ within a given set $\{a_{i,N}\}$, and $h_{\{a_{i,N}\}}$ are rational coefficients for each of these sets. Explicit results for $N = 5, 6, 7$ are

$$\mathcal{H}_5^{\mathrm{nl}} = -\tfrac{5}{36}\,\mathcal{J}'^4 + \tfrac{5}{63}\,\mathcal{J}''^3\,, \tag{131}$$

$$\mathcal{H}_6^{\mathrm{nl}} = -\tfrac{5}{6}\,\mathcal{J}\mathcal{J}'^4 + \tfrac{10}{21}\,\mathcal{J}\mathcal{J}''^3 + \tfrac{1}{2}\,\mathcal{J}'^2\mathcal{J}''^2 - \tfrac{5}{42}\,\mathcal{J}''\mathcal{J}'''^2\,, \tag{132}$$

$$\mathcal{H}_7^{\mathrm{nl}} = -\tfrac{35}{12}\,\mathcal{J}^2\mathcal{J}'^4 + \tfrac{5}{3}\,\mathcal{J}^2\mathcal{J}''^3 + \tfrac{7}{2}\,\mathcal{J}\mathcal{J}'^2\mathcal{J}''^2 - \tfrac{5}{6}\,\mathcal{J}\mathcal{J}''\mathcal{J}'''^2$$
$$+ \tfrac{7}{24}\,\mathcal{J}'^4 - \tfrac{1}{4}\,\mathcal{J}'^2\mathcal{J}'''^2 + \tfrac{7}{132}\,\mathcal{J}''\big(\mathcal{J}^{(4)}\big)^2\,. \tag{133}$$

The split in (126) into $N\,\mathcal{H}_{N-1}^{\mathrm{nl}}$ and $\partial_\sigma R_N$ is ambiguous, since we can add total derivative terms to the former and subtract them from the latter. Above we made particular choices for $\mathcal{H}_N^{\mathrm{nl}}$ that minimize the number of derivatives acting on the $\mathcal{J}$-factor with the highest number of derivatives for each term. (For instance, the term $\mathcal{J}'^2\mathcal{J}'''^2$ in (133), up to total derivative terms and a term proportional to $\mathcal{J}''^4$, is equivalent to $-\mathcal{J}'^2\mathcal{J}''\mathcal{J}^{(4)}$, which has four derivatives on the last factor, whereas the original term had at most three derivatives.)

A cross-check on the numerical factors in (121)-(126) is the relation [61]

$$\frac{\partial R_N}{\partial \mathcal{J}} = N\,R_{N-1}\,. \tag{134}$$

Note that additionally all black terms in (126) (the ones displayed explicitly) are mapped to each other via (134); the same is true for the red and blue terms for our choice of $\mathcal{H}_N^{\mathrm{nl}}$.

The associated Hamiltonian densities $\mathcal{H}_N$ obeying

$$\delta\mathcal{H}_N \sim R_N\,\delta\mathcal{J} \sim \tfrac{1}{N+1}\,\delta R_{N+1}\,, \tag{135}$$

(where $\sim$ denotes equality up to total derivative terms) are given by

$$\mathcal{H}_0 = \mathcal{J}\,, \qquad \mathcal{H}_1 = \tfrac{1}{2}\,\mathcal{J}^2\,, \qquad \mathcal{H}_2 = \tfrac{1}{3}\,\mathcal{J}^3 - \tfrac{1}{6}\,\mathcal{J}'^2\,, \tag{136}$$

$$\mathcal{H}_3 = \tfrac{1}{4}\,\mathcal{J}^4 - \tfrac{1}{2}\,\mathcal{J}\mathcal{J}'^2 + \tfrac{1}{20}\,\mathcal{J}''^2\,, \tag{137}$$

$$\mathcal{H}_4 = \tfrac{1}{5}\,\mathcal{J}^5 - \mathcal{J}^2\mathcal{J}'^2 + \tfrac{1}{5}\,\mathcal{J}\mathcal{J}''^2 - \tfrac{1}{70}\,\mathcal{J}'''^2\,, \tag{138}$$

$$\mathcal{H}_5 = \tfrac{1}{6}\,\mathcal{J}^6 - \tfrac{5}{3}\,\mathcal{J}^3\mathcal{J}'^2 + \tfrac{1}{2}\,\mathcal{J}^2\mathcal{J}''^2 - \tfrac{1}{14}\,\mathcal{J}\mathcal{J}'''^2 + \tfrac{1}{252}\,\big(\mathcal{J}^{(4)}\big)^2 + \mathcal{H}_5^{\mathrm{nl}}\,, \tag{139}$$

$$\mathcal{H}_6 = \tfrac{1}{7}\,\mathcal{J}^7 - \tfrac{5}{2}\,\mathcal{J}^4\mathcal{J}'^2 + \mathcal{J}^3\mathcal{J}''^2 - \tfrac{3}{14}\,\mathcal{J}^2\mathcal{J}'''^2 + \tfrac{1}{42}\,\mathcal{J}\big(\mathcal{J}^{(4)}\big)^2 - \tfrac{1}{924}\,\big(\mathcal{J}^{(5)}\big)^2 + \mathcal{H}_6^{\mathrm{nl}}\,, \tag{140}$$

$$\vdots = \vdots$$

$$\mathcal{H}_N = \tfrac{1}{N+1}\,\mathcal{J}^{N+1} + \sum_{i=1}^{N-1}(-1)^i\,\frac{((i+1)!)^2}{(2i+2)!}\binom{N}{i+1}\,\mathcal{J}^{N-i-1}\big(\partial_\sigma^i\mathcal{J}\big)^2 + \mathcal{H}_N^{\mathrm{nl}}\,. \tag{141}$$

Inspection of the explicit results above shows compatibility with (80), which is a simple, yet non-trivial, cross-check on the correctness of the equations displayed in this appendix.

Starting with the general form of the Hamiltonian density (141), analytically continuing in $N \leq 1$, assuming $N = \varepsilon \to 0^+$, inserting $\mathcal{J} = \Phi'$ and dropping total derivative terms yields the limiting Hamiltonian density

$$\mathcal{H}_\varepsilon \sim \varepsilon\,\Phi'\ln\Phi' + \mathcal{O}(\varepsilon^2)\,, \tag{142}$$

which remains finite and non-trivial if rescaled by $1/\varepsilon$.

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
