# Peer review of "Near horizon dynamics of three dimensional black holes"

_SciPost Physics, doi:SciPost Phys. 8, 010 (2020)_

## Round 2 · Referee Report · Anonymous (Referee 1) · 2019-9-5

Report

Let me address all of the points here.

The present article builds upon the recent “fluff proposal” of one of the authors for a description of BTZ microstates. Broadly speaking, the goal of that and this work is to obtain a quantum theory from the gravitational path integral on BTZ spacetimes whose microcanonical entropy reproduces the standard BTZ entropy.

The authors’ approach is to start with the Chern-Simons formulation of AdS3 gravity and to try to obtain a path integral description of the BTZ horizon, in terms of the boundary modes on a “stretched” horizon (although it seems that one can take the stretched horizon all the way to the actual horizon on account of the topological nature of 3d gravity). This procedure depends sensitively on the boundary conditions one imposes on this stretched horizon, which in turn leads to a chiral WZW like theory subject to constraints and with an effective Hamiltonian determined by these boundary conditions.

Strengths:

  1. The present work is timely and of significant interest. The question of BTZ microstate counting is an old and important one of course, but the present approach also touches on the question of whether or not soft hair can account for black hole entropy (with the modes on the stretched horizon playing the role of the soft hair), and further the path integrals landed upon are, in a moral sense, 2d cousins of the Schwarzian theory of so much recent interest.

  2. Because the authors start from the Chern-Simons description of 3d gravity, the “rules of the game” are well-established and well-defined. There are the usual statements about how the CS and metric descriptions of gravity differ non-perturbatively (and perhaps even perturbatively, when it comes to the field range of gravitational zero modes), from the point of view of computing the classical phase space and loop effects around a background the CS description is perfectly adequate.

Weaknesses:

  1. Unfortunately I do not think that the analysis undertaken in this paper is correct. It is missing pieces. The important point is that the authors consider the path integral description of 3d gravity in the exterior region of a BTZ black hole. This region is topologically of the form annulus times time, with one boundary of the annulus being the stretched horizon and the other being conformal infinity. Now the point is that when quantizing Chern-Simons theory on the annulus (times time) according to the “constrain first” approach of first integrating out $A_0$, one has a residual integral over flat connections on the annulus. These flat connections are parameterized as $A_i = G^{-1} \partial_i G$ with $i$ indexing spatial directions. Without yet accounting for the current constraints imposed by boundary conditions, one then finds a theory living on the boundary of the annulus with fundamental fields $G|_1$ and $G|_2$, the boundary values of $G$ on the two boundaries, and the monodromy which can be non-trivial and which the authors label as $b$ in their work. While the boundary values of $G$ can depend on the boundary directions, $b$ can depend on time, and indeed the full path integral is over all three fields. (Semiclassically in CS theory with gauge group $G$ on the annulus, the integral over $b$ imposes the constraint that the $G$-valued charges on one boundary are the same on the other. While the fields $G|_1$ and $G|_2$ do not directly couple to each other, they do couple to $b$, which leads to these charge relations.)

(By the by, the authors desire to work around a fixed black hole, which corresponds to a fixed $b$. I think this can be achieved in the full path integral by imposing that $b$ asymptotes in the infinite past and future to the desired value. But one still has the fluctuations of $b$ to deal with.)

In the authors approach they set by hand $b$ to a constant and neglect the $G$ on the boundary at conformal infinity. Without fixing this, it is unclear whether the conclusions reached by the authors, like the current descendants of the vacuum of the near-horizon theory carrying no energy, are reliable.

It could very well be that the authors’ conclusions about the horizon modes are unaltered when including the other fields, but that should be the output of a computation.

  1. Furthermore the authors tend to follow the approach of Coussaert, Henneaux, and van Driel from their landmark paper showing that AdS3 gravity is equipped with Liouville theory on the boundary. However, their approach, while morally correct, makes several obvious mistakes at various points even at the level of classical physics, and so one should be careful. For example one claim of this old paper is that the $SL(2;R)\times SL(2;R)$ Chern-Simons theory on the Lorentzian cylinder leads to non-chiral $SL(2;R)$ WZW on the boundary. But this is obviously not true. $G$ Chern-Simons theory on the cylinder always gives $G$ chiral WZW on the boundary. In other words, from AdS3 gravity on global AdS one finds current algebra, not the full WZW model, including vertex operators.

From the point of view of the present article, the thing I would like to see is a careful accounting of the redundancies (or screening charges) and boundary conditions that arise in their quantization. For example, in the old Coussaert/Henneaux/van Driel analysis, well, we know that the phase space of AdS3 gravity is two copies of the quotient space Diff$(\mathbb{S}^1)/PSL(2;\mathbb{R})$. The boundary model one obtains should have the same phase space. The same applies here for the near-horizon model: upon accounting for the redundancies, the phase space of the authors’ model should coincide with that of the near-horizon region.

—-

I also have a question about a point I find confusing in the authors’ analysis. The authors propose various boundary conditions at the horizon, and then work out the Hamiltonian that follows. However, shouldn’t it be the case that it is not up to us what boundary conditions to choose? For example, if we were interested in computing fully retarded functions, then one ought to impose mere regularity on the future horizon, as one does in the fluid/gravity correspondence.

Recommendation:

At this time I cannot recommend publication, largely on account of point (1) described above.

  • validity: good
  • significance: good
  • originality: good
  • clarity: high
  • formatting: excellent
  • grammar: excellent

Author:  Daniel Grumiller  on 2019-09-27  [id 610]

(in reply to Report 1 on 2019-09-05)
Category:
correction

** Response to referee 1 **

We thank the referee for a careful reading of the manuscript and their extensive comments. In response to these comments we realize we should clarify a few things about our overall approach. We start with replies to the issues raised in the report and then list the changes implemented in the manuscript.

** Issue 1 **

In our work we discuss the reduction of Chern-Simons theory to a boundary theory for manifolds which are topologically the (punctured) disk times time. Thus, the referee is correct to point out that we consider topologies of the form annulus times time rather than a cylinder. We corrected the statements in subsection 2.2 accordingly.

However, the reason that in our work we only consider one boundary component, namely a stretched horizon, is that at the other (asymptotic) boundary the same boundary conditions are induced, which is explicit in our starting point (13), since the state-dependent information of the Chern-Simons connection manifestly is independent from the radial coordinate. Hence, also all physical observables - holonomies and boundary charges - are independent from the radial coordinate.

We emphasize that [as stated in section 2, below Eq. (12)] imposing boundary conditions is an important part of the specification of the theory, and we do have a choice here. We could take them, for instance, to be Brown-Henneaux, Compere-Song-Strominger, or of near-horizon type. So a priori there is nothing that forces us to pick a specific set of boundary conditions at either of the boundaries (as long as they are internally consistent and consistent with each other). Our choice to pick near horizon boundary condition is indeed consistent.

Note that the terminology 'near-horizon' can be misleading, because we work in a gauge where the radial dependence is gauged away. Hence we can effectively impose the 'near-horizon' boundary conditions asymptotically just like in the usual Brown-Henneaux story. From the Chern-Simons perspective this is equivalent to demanding the Chern-Simons connections to be in the hyperbolic conjugacy class of SL(2), which is a statement that does not depend on the radial dependence of the connection.

Having said all this, we would like to defend our choice to label the boundary conditions as 'near horizon': first of all, and perhaps most convincingly, they were discovered in Ref. [22] by starting from a near horizon expansion; second, the properties reviewed at the end of section 2.3 show several unique features associated with these boundary conditions so that it seems appropriate to name them; in particular, the regularity of all excitations (unlike any other boundary conditions) makes these boundary conditions tailor-made for a near horizon discussion. So despite of potential confusion that our terminology may create we think it is appropriate to stick to 'near horizon boundary conditions'.

To address the points above we have made suitable modifications and added explanations in the manuscript, see below for a list of changes.

** Issue 2 **

Regarding the second point of the referee, we note that, unlike the Coussaert-Henneaux-van Driel analysis, we do not combine the two chiral sectors and indeed find the two chiral WZW theories in Eq. (30), one for each chiral sector (as denoted in the first paragraph of section 3.1, we only consider one chiral half of the theory in section 3). We furthermore already show that the phase space of the boundary theory [with Poisson brackets in Eq. (52)] coincides with the analysis from the near-horizon boundary conditions [given in Eq. (18)]. We are thus not sure what the referee is proposing us to do further and have not made any changes in the manuscript regarding this issue.

** Further comments by the referee **

There were additional, by comparison minor, statements made by the referee that we would like to address here: The statement that b can depend on time, while correct, seems irrelevant for us; we assume that db/dt vanishes at the boundary and that the holonomy is time-independent; this is certainly true for the BTZ black hole and all its soft hairy excitations, as our discussion of the reducibility parameter in section 2.3 shows. Since we address the time-dependence of b already briefly in footnote 3 we did not make additional changes.

Regarding the final question, we are convinced that it is possible to choose various sets of (consistent) boundary conditions - like Brown-Henneaux, Compere-Song-Strominger or near-horizon - and that it depends on the precise physical question one wants to address which of these boundary conditions should be imposed.

** Changes in the manuscript **

In order to clarify issue 1, we made the following changes/additions to the manuscript:

  1. We comment further on the topology of the manifold that we are using (correcting the statement about which topology we use). The new text is contained in the first two paragraphs of subsection 2.2, which replace the previous first paragraph.

  2. We emphasize in the last (new) paragraph of section 2.3 the fact that our `near-horizon' boundary conditions can be imposed asymptotically and further highlight the list of properties that justify this nomenclature.

** Response to referee 2 **

We thank the referee for a careful reading of the manuscript and their comments. In response to the requested changes we have made the following minor edits to our manuscript

  1. In the lines below (16) we have replaced the ... by arbitrary gauge parameters followed by the remark that these gauge parameters do not appear in the transformation law or in the charges below. This also implies that these terms do not affect the result in Eq. (40).

  2. We have added a footnote in the first line of section 2.1 clarifying that the relation between the Chern-Simons and metric formulations of 3D gravity is not fully understood at the quantum level.

  3. In order to counteract the epsilon-proliferation in this paper we have changed notation in and around Eqs. (62)-(63) to xi and in section 5 and 6 we have replaced epsilon in different font. We hope this clarifies the presentation.

---

## Round 2 · Referee Report · Anonymous (Referee 2) · 2019-9-13

Strengths

1- A novel way to generate a cutoff for the soft hair of black holes 2- a potential clue to resolve the tension between finite black hole entropy vs infinite degeneracy of soft hair 3- nice way of organizing boundary conditions for 3d Einstein gravity 4-Written in a well-organized and self-contained way, with useful comparison to interesting related work. Very pleasant to read.

Weaknesses

1- doesn't provide a way to derive the coefficient in eq. (103)

Report

This paper considers the Hamiltonian reduction of $SL(2,R)\times SL(2,R)$ Chern-Simons theory, which is equivalent to Einstein gravity with negative cosmological constant in 3d at the classical level. The resulting Hamiltonian depends on the boundary conditions. The authors constructs a one-parameter family of such Hamiltonians.
By doing so, this paper tries to address the question how the black holes entropy can be reproduced from the soft modes, given the infinite degeneracy of the later. By analytically continuing the aforementioned parameter, the authors introduce a cutoff for the soft hair, which is potentially useful for understanding the micro states of black holes.

I recommend the paper to published if the authors consider the following comments in the section below.

Requested changes

1-What does $\cdots$ in the line below (16) denote? If it contains $L_\pm$ components, the transformation $\delta a_\sigma=\partial_\sigma \epsilon+[a_\sigma,\epsilon]$ will contain terms proportional to $L_\pm$ as well. If this is true, this seems to affect the result eq. (40).

2-At the beginning of section 2.1, it will be useful to point out that the relation between Chern-Simons action and Einstein gravity is not fully understood at the quantum level.

3-It seems that the same notation $\epsilon$ has different meanings in various places. In eq. (14) and the line below eq. (16) $\epsilon$ denotes the asymptotic symmetry transformation, and is matrix-valued. In eq. (62) and eq. (63) $\epsilon$ is a function. Finally in section 5, $\epsilon$ denotes a number between 0 and 1. I think it will be helpful to distinguish them.

---

## Round 3 · Referee Report · Anonymous · 2019-10-23

Report

I recommend the revised manuscript to be published.

---

## Round 3 · List of Changes

1. In the lines below (16) we have replaced the ... by arbitrary gauge parameters followed by the remark that these gauge parameters do not appear in the transformation law or in the charges.

2. Added a footnote in the first line of subsection 2.1.

3. The first two paragraphs of subsection 2.2 are new and replace/correct the previous first paragraph.

4. The last paragraph of subsection 2.3 is new.

5. Added third sentence in beginning of section 3.

6. Changed notation in and around Eqs. (62)-(63) from epsilon to xi to distinguish it from epsilon in other sections.

7. In sections 5 and 6 we changed to font for epsilon to distinguish it from epsilon in previous sections.

---

## Editorial Decision

published